# *Accept–reject* decision-making revealed via a quantitative and ethological study of *C. elegans* foraging

**Jessica A Haley[1,2], Tianyi Chen[2], Mikio Aoi[3,4]\*, Sreekanth H Chalasani[2]\***

[1]Neurosciences Graduate Program, University of California, San Diego, San Diego, United States; [2]Molecular Neurobiology Laboratory, Salk Institute for Biological Studies, La Jolla, United States; [3]Halıcıoğlu Data Science Institute, University of California, San Diego, La Jolla, United States; [4]Department of Neurobiology, University of California, San Diego, La Jolla, United States

**\*For correspondence:**
maoi@ucsd.edu (MA);
schalasani@salk.edu (SHC)

**Competing interest:** The authors declare that no competing interests exist.

## eLife Assessment

Understanding how neural circuits mediate decision-making is a core problem in neuroscience. In this interesting and **important** work, the authors use detailed behavioral analysis and rigorous quantitative modeling to **convincingly** support the idea that the nematode *C. elegans* uses an "accept-reject" behavioral strategy, based on learned features of its environment, to make decisions upon encountering food patches. The work expands our understanding of the behavioral repertoire of this species, providing a foundation for future mechanistic studies in this powerful model system.

**Abstract** Decision-making is a ubiquitous component of animal behavior that is often studied in the context of foraging. Foragers make a series of decisions while locating food (food search), choosing between food types (diet or patch choice), and allocating time spent within patches of food (patch-leaving). Here, we introduce a framework for investigating foraging decisions using detailed analysis of individual behavior and quantitative modeling in the nematode *Caenorhabditis elegans*. We demonstrate that *C. elegans* make *accept–reject* patch choice decisions upon encounter with food. Specifically, we show that when foraging among small, dispersed, and dilute patches of bacteria, *C. elegans* initially *reject* several bacterial patches, opting to prioritize exploration of the environment, before switching to a more exploitatory foraging strategy during subsequent encounters. Observed across a range of bacterial patch densities, sizes, and distributions, we use a quantitative model to show that this decision to *explore* or *exploit* is guided by available sensory information, internal satiety signals, and learned environmental statistics related to the bacterial density of recently encountered and *exploited* patches. We behaviorally validated model predictions on animals that had been food-deprived, animals foraging in environments with multiple patch densities, and null mutants with defective sensory modalities. Broadly, we present a framework to study ecologically relevant foraging decisions that could guide future investigations into the cellular and molecular mechanisms underlying decision-making.

## Introduction

Decision-making is frequently defined as the selection of a course of action among several alternatives. The ubiquity and importance of decision-making has led to its use in describing a broad range of behaviors from taxes of unicellular organisms to economics and politics in human society (*Lee, 2013*; *Budaev et al., 2019*). In decision neuroscience, laboratory experiments have investigated the

behavioral choices of animals in well-controlled environments or tasks (*Mobbs et al., 2018*). While these experiments have led to key insights into mechanisms underlying decision-making and related cognitive processes (*Gold and Shadlen, 2007*), a comprehensive understanding of decision-making remains elusive. Many researchers have advocated for a more neuroethological approach – suggesting that experiments designed to understand problems the brain evolved to solve offer a more rigorous framework for investigation (*Mobbs et al., 2018*; *Stephens, 2008*; *Hall McMaster and Luyckx, 2019*). Thus, one approach to enrich our understanding of decision-making is to look at the decisions made by animals foraging in naturalistic environments.

Foraging animals make a hierarchy of decisions to locate food (food search), choose between different food types (diet or patch choice), and allocate time spent within patches of food items (patch-leaving) (*Stephens, 2008*; *Schoener, 1971*; *Pyke et al., 1977*; *Stephens and Krebs, 1986*; *Stephens et al., 2007*). These foraging decisions often require that an animal *exploit* an environment for known resources or *explore* it for potentially better opportunities elsewhere (*Stephens and Krebs, 1986*; *Nonacs, 2001*). This *exploration–exploitation* trade-off requires cognitive computations such as learning the spatiotemporal distribution of food, route planning, estimation of food availability, and decision-making (*Hills, 2006*; *Calhoun and Hayden, 2015*). According to optimal foraging theory, foragers may seek to maximize their rate of net energy gained over time (*Pyke et al., 1977*; *Stephens and Krebs, 1986*; *Charnov, 1976*) by using internal and external information to guide decision-making, especially in environments where resources are sparsely distributed or fluctuating. Although many studies have provided insight into the motivations, behavioral implementations, and genes associated with foraging decisions (*Stephens and Krebs, 1986*; *Stephens et al., 2007*), a more rigorous framework for exploring the neuronal mechanisms underlying foraging decisions is essential to establish a comprehensive understanding of decision-making.

The microscopic nematode *Caenorhabditis elegans* is well suited for investigating the cellular and molecular basis of foraging decisions (*Haley and Chalasani, 2024*). A myriad of genetic tools, behavioral assays, and neuronal imaging techniques have been developed to take advantage of the species' quick reproductive cycle, isogeneity, optical transparency, and ease of maintenance (*Brenner, 1974*; *Faumont and Lockery, 2006*; *Boulin and Hobert, 2012*). While *C. elegans* typically feed upon a diversity of bacterial types in the wild (*Samuel et al., 2016*), they are commonly maintained in the laboratory on agar plates containing large patches of the bacteria *Escherichia coli* as a food source (*Brenner, 1974*). Even in these simplified laboratory conditions and despite having a numerically simple nervous system of only 302 neurons (*Brenner, 1974*; *White et al., 1986*), *C. elegans* display complex and robust species-typical behaviors (*de Bono and Maricq, 2005*) involving learning and memory (*Colbert and Bargmann, 1995*; *Zhang et al., 2005*; *Ardiel and Rankin, 2010*; *Calhoun et al., 2015*), and decision-making (*Bendesky et al., 2011*; *Faumont et al., 2012*; *Iwanir et al., 2016*; *Tanimoto et al., 2017*; *Ji et al., 2021*; *Scheer and Bargmann, 2023*). Recent studies have demonstrated that ecologically focused environmental enrichment permits identification of novel behaviors and gene functions in *C. elegans* and other animals (*Petersen et al., 2015*; *Volgin et al., 2018*; *Kempermann, 2019*; *Guisnet et al., 2021*).

Foraging animals often must make one of two types of decisions: *stay–switch* or *accept–reject*. *Stay–switch* decisions refer to scenarios where an individual experiences diminishing returns with the current action and must decide when to switch to a new action (e.g., patch-leaving and area-restricted search) (*Stephens, 2008*; *Schoener, 1971*; *Pyke et al., 1977*; *Stephens and Krebs, 1986*; *Stephens et al., 2007*). In contrast, *accept–reject* decisions refer to situations where an individual must decide between engaging with an option or ignoring it in search of a better one (e.g., diet or patch choice) (*Stephens, 2008*; *Pyke et al., 1977*; *Stephens and Krebs, 1986*). *Stay–switch* decisions have been well described in *C. elegans* (*Calhoun et al., 2015*; *Bendesky et al., 2011*; *Shtonda and Avery, 2006*; *Hills et al., 2004*; *Calhoun et al., 2014*; *Gray et al., 2005*; *Milward et al., 2011*; *Olofsson, 2014*) and enable them to successfully explore an environment and exploit the bacteria within (*Iwanir et al., 2016*; *Pradhan et al., 2019*; *Gloria-Soria and Azevedo, 2008*; *Madirolas et al., 2023*). However, to our knowledge, *accept–reject* decisions have not yet been demonstrated in *C. elegans*. While *C. elegans* have been shown to alter food preferences in a diet choice assay, this behavior has only been described by a set of *stay–switch* decisions (*Scheer and Bargmann, 2023*; *Shtonda and Avery, 2006*; *Madirolas et al., 2023*). The ability to make *accept–reject* decisions is likely advantageous for foraging in fluctuating or variable environments as rejecting an encountered food item of low quality

creates the opportunity for a subsequent encounter with preferred food. Thus, while *C. elegans* can successfully forage in patchily distributed environments, it is not known whether *C. elegans* make decisions to *exploit* a patch of food upon encounter and how this decision-making process changes over a series of patch encounters.

In this study, we aimed to identify if *C. elegans* make *accept–reject* decisions upon encounter with bacterial patches. To answer this question, we performed a detailed analysis of the behavior of individual animals foraging in an ecologically inspired environment where bacterial patches were dilute and dispersed. In conditions where the bacterial density was much lower than that of previously visited patches, animals first explored the environment before *accepting* patches for exploitation. This explore-then-exploit strategy persists even when only one bacterial patch is present in the environment. Using a theoretical framework, we found that this initial exploration reflects a series of *accept–reject* decisions guided by signals related to the bacterial density of current and recently *explored* and *exploited* patches as well as the animal's state of satiety.

## Results

### *C. elegans* forage in patchy environments with an explore-then-exploit strategy

To investigate whether *C. elegans* make *accept–reject* decisions upon encounter with bacterial patches, we observed the behavior of animals foraging on an agar surface containing an isometric grid of small, low-density bacterial patches (*Figure 1A*, *Videos 1 and 2*). Animals were confined to an arena where their behavior was recorded and tracked for 60 min (*Figure 1A*, *Figure 1—figure supplement 1*, *Video 3*). During this time, animals were observed to move about the arena (*Figure 1A–C*, *Figure 1—figure supplement 2*, *Video 4*) with each animal encountering, on average, ~8.4 total patches and ~5.3 unique patches during the 1-hr recording (*Figure 1D*). These patch encounters ranged in duration from seconds to tens of minutes and could be classified as either short or long (2+ min) using a Gaussian mixture model (GMM) (*Figure 1E*, *Figure 1—figure supplement 3*). Notably, animals were significantly more likely to stay on patch for longer durations at later time points (*Figure 1F*), which corresponded to an overall increase in their probability of residing on patch (*Figure 1G*, *Figure 1—figure supplement 4*). Specifically, we found that while animals initially resided on bacterial patches at levels close to those predicted by chance, an animal's probability of residing on a bacterial patch increased significantly after ~10 min. These findings, that patch duration and residence increased over time, suggest that animals are more likely to exploit an encountered patch as time goes on.

Given that the bacterial density of the experimental patches was approximately 20 times more dilute than that of conditions animals experienced during development and immediately preceding the assay, it was possible that the initial delay in exploitation could be due to animals' inability to detect the presence of bacteria in these more dilute patches. We therefore sought to determine if animals detected patches at all time points. Previous studies have shown that *C. elegans* display an immediate and marked slowdown upon encounter with the edge of a food patch and sustain these slower speeds on food (*Iwanir et al., 2016*; *Fujiwara et al., 2002*). Therefore, velocity can be used as a proxy for an animal's ability to sense a bacterial patch. Following this precedent, we quantified the instantaneous velocity of individuals in our assay (*Figure 1H*). We observed significant deceleration of an animal's instantaneous velocity upon encounter with the patch edge (*Figure 1I*), with no obvious difference between short-duration, early time point (*Figure 1J*) and long-duration, late time point (*Figure 1K*) encounters. Animals consistently displayed marked slowdown when approaching the patch edge (*Figure 1L*), achieving an average deceleration of –19.5 µm/s$^2$ upon encounter with the bacterial patch (*Figure 1M*, *Figure 1—figure supplement 5*). This slowdown was statistically significant for both long- and short-duration encounters (*Figure 1M*). Further, consistent with *C. elegans* behavior on larger and more densely seeded patches in other studies (*Sawin et al., 2000*), animals in our assay maintained significantly lower average velocities on patch (~52 µm/s) compared to off patch (~198 µm/s) (*Figure 1N, O*). These results suggest that, despite detecting the availability of food during early patch encounters, *C. elegans* opt to initially prioritize exploration of the environment before switching to a more exploitatory foraging strategy during subsequent patch encounters.

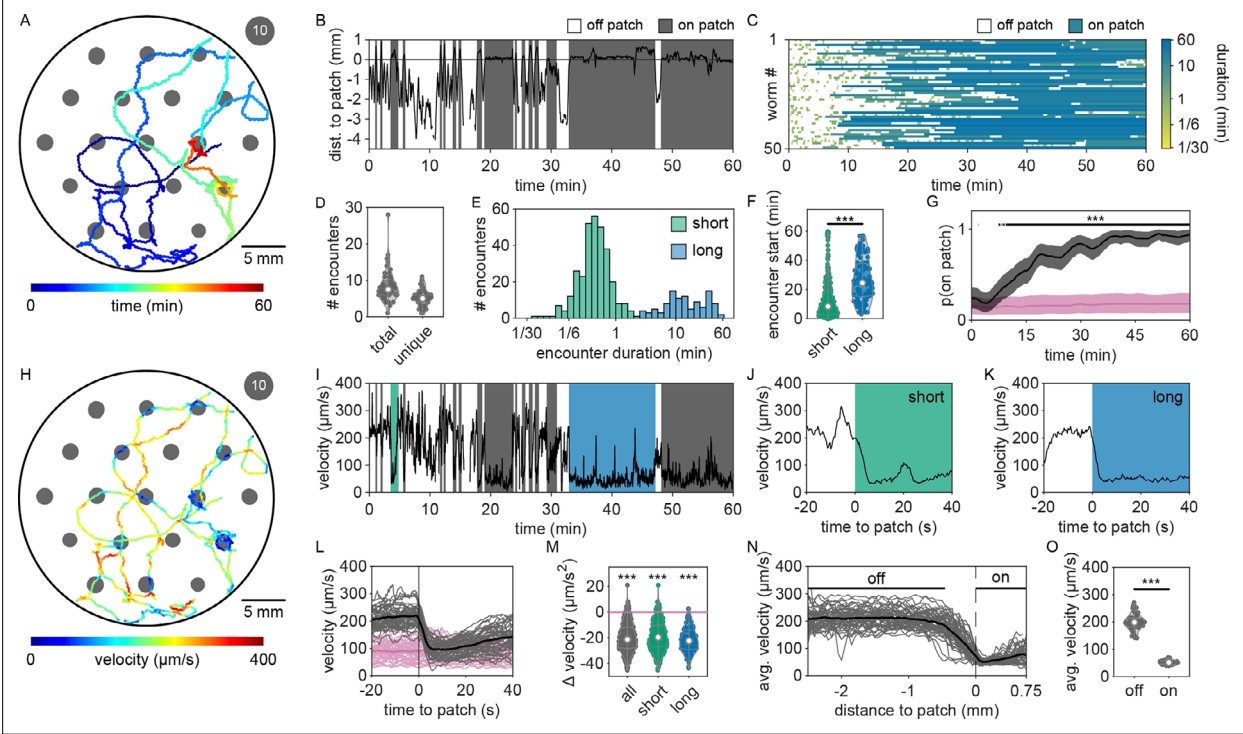

**Figure 1.** *C. elegans* forage in a patchy environment with an explore-then-exploit strategy. (**A**) An example animal's midbody location (colored to represent time in the experiment) as it forages in an environment bounded by a large (30 mm diameter) arena containing 19 small (~1.8 mm diameter) bacterial patches (gray). Each patch was made by pipetting ~0.5 µl of OP50 *E. coli* diluted to $OD_{600}$ ~10 and grown at room temperature for ~1 hr. (**B**) Distance between the example animal's midbody position and the nearest patch edge (positive indicates inside patch) is plotted (black) for every time point. Putative encounters with a bacterial patch are indicated (gray). (**C**) Patch encounters (colored to represent the duration of the encounter) for 50 individuals foraging in these environments are plotted. (**D**) The number of total and unique patch encounters for each animal is shown. (**E**) Duration for each patch encounter was computed and classified as either short (0–2 min) or long (2–60 min) using a Gaussian mixture model. The distributions of all observed short and long encounters are plotted with duration binned logarithmically. (**F**) The observed start time of each patch encounter is shown for all short- and long-duration encounters. Long-duration encounters occur significantly later (one-tailed Mann–Whitney *U*-test). (**G**) The probability of residing on patch was computed for all worms across time (black) and compared to the probability of residing on patch if patch locations were semi-randomly permuted (pink). Smoothed median values are plotted with bootstrap-derived 2.5% and 97.5% quantiles shown in shaded regions. Time points where observed probabilities of residing on patch significantly exceed permuted probabilities are indicated by a black line (one-tailed Fisher's exact test with Benjamini–Hochberg correction). (**H**) The track of the example animal in (**A**) is replotted with color used to represent the animal's instantaneous velocity at each time point. (**I**) Velocity of the example animal over time is plotted (black) alongside patch encounters (gray) as previously identified in (**B**). Example encounters – one early, short duration (green) and one late, long duration (blue) – are indicated. (**J**, **K**) A 60-s time window surrounding the start of these example encounters is enlarged. (**L**) Velocity trajectories were aligned to patch entry for every encounter. Mean encounter-aligned (black/gray) and randomly aligned (pink) trajectories are plotted for each animal (light) and across all animals (dark). (**M**) Deceleration upon encounter with the patch edge is plotted for every encounter and grouped by duration type. Deceleration was significantly lower for encounter-aligned as compared to randomly aligned trajectories (pink) for all duration types (one-tailed Mann–Whitney *U*-tests with Bonferroni correction). (**N**) Mean velocity as a function of the distance from the edge of bacterial patches (computed for 50 µm bins) is shown for every animal (gray) and across all animals (black). (**O**) Each animal's mean velocity during time spent on and off (midpoint at least –0.46 mm from patch edge) patch is shown. Velocity on patch was significantly slower than off patch (one-tailed paired-sample *t*-test). Sample data for one animal (worm #1) are shown in (**A–C**, **H–K**). Summary data for all animals (*N* = 50 worms) and encounters (*N* = 419 total encounters) are shown in (**D–G**, **L–O**). Violin plots in (**D**, **F**, **M**, **O**) give the kernel density estimate (KDE) and quartiles for each measure. Asterisks denote statistical significance (***p < 0.001). See also *Figure 1—figure supplements 1–5* and *Videos 1–4*.

The online version of this article includes the following source data and figure supplement(s) for figure 1:

**Source data 1.** Number of total and unique patch encounters in *Figure 1D*.

**Source data 2.** Duration of patch encounters in *Figure 1E*.

**Source data 3.** Start time of patch encounters in *Figure 1F*.

**Source data 4.** On-patch residence in *Figure 1G*.

**Source data 5.** Velocity of animals upon encounter with patch edge in *Figure 1L*.

**Source data 6.** Deceleration of animals upon encounter with patch edge in *Figure 1M*.

*Figure 1 continued on next page*

*Figure 1 continued*

**Source data 7.** Velocity of animals upon encounter with patch edge in *Figure 1N*.

**Source data 8.** Velocity on and off patch in *Figure 1O*.

**Figure supplement 1.** Assay preparation, arena and patch detection, and behavioral tracking.

**Figure supplement 2.** Defining a patch encounter using high-resolution behavioral recordings.

**Figure supplement 3.** Classifying encounters based on duration using a Gaussian mixture model (GMM).

**Figure supplement 4.** Permuting patches to test for significance of the observed time-dependent increase in patch residence.

**Figure supplement 5.** Analyzing deceleration upon encounter with a patch.

## The timing of the switch from exploration to exploitation varies with bacterial density

To determine whether this explore-then-exploit foraging strategy is dependent upon food-related characteristics of the environment, we varied the density of the bacterial patches. By diluting bacterial stocks and controlling growth time, we created 12 bacterial density conditions with relative density ranging from 0 to ~200 (*Figure 2A*, *Figure 2—figure supplements 1–3*). At the low end of this range (densities less than ~0.5), animals do not appear to detect bacterial patches as indicated by on-patch velocities matching those of animals foraging on bacteria-free (density 0) patches (*Figure 2B, C*). At the high end of the range (density ~200), bacterial density is comparable to the environments animals experienced during development and immediately prior to the assay (*Figure 2—figure supplements 2 and 3*).

Previous studies have shown that in diet choice assays where animals are given the option between patches of varying quality or density, animals spend more time on preferred patches (*Scheer and Bargmann, 2023*; *Shtonda and Avery, 2006*; *Madirolas et al., 2023*). We hypothesized that even when all patches in the environment are of equal density, an efficient forager should still modulate the amount of time spent on the bacterial patches in a density-dependent manner. To test this, behavior was recorded and tracked for 1 hr as animals foraged in patchy environments matching one of the bacterial density conditions (*Figure 2D*, *Figure 2—figure supplement 4*). Consistent with our hypothesis, we found that the total time animals spent on bacterial patches increased monotonically with increasing bacterial density following a sigmoidal trend (*Figure 2E*). Further, the probability of an animal residing on patch as a function of time was highly dependent on the bacterial density (*Figure 2F*, *Figure 2—figure supplement 5*). At the highest densities (50 and 200), animals

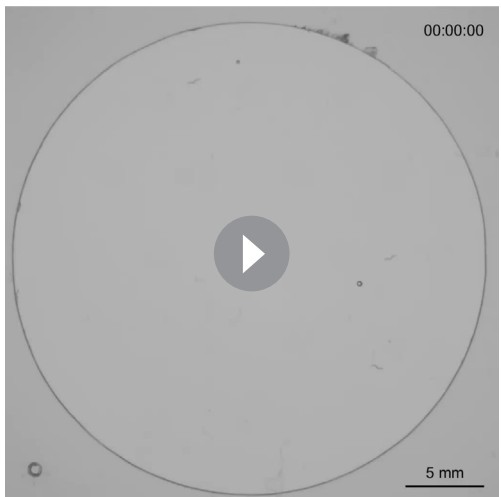

**Video 1.** *C. elegans* foraging in a patchily distributed environment. Four adult *C. elegans* forage for 1 hr in a 30-mm arena containing an isometric grid of small, low-density bacterial patches.

https://elifesciences.org/articles/103191/figures#video1

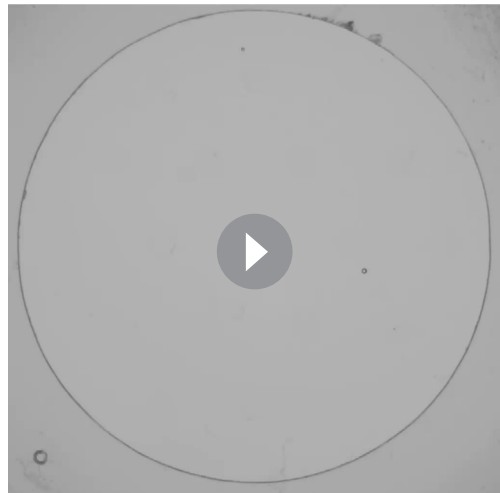

**Video 2.** Visualizing dilute bacterial patches using diffraction of light. A piece of dark cardstock passed between the light source and the assay plate enables visualization of dilute bacterial patches.

https://elifesciences.org/articles/103191/figures#video2

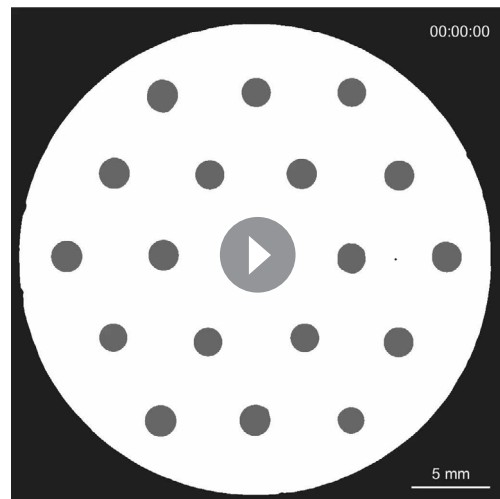

**Video 3.** Tracking the midbody location of an animal foraging in a patchily distributed environment. The midbody location of one of the four animals shown in *Video 1* is shown as the animal forages for 1 hr in a 30-mm arena containing an isometric grid of small, low-density bacterial patches. The animal shown corresponds to the example animal observed in *Figure 1A, B, H–K*.

https://elifesciences.org/articles/103191/figures#video3

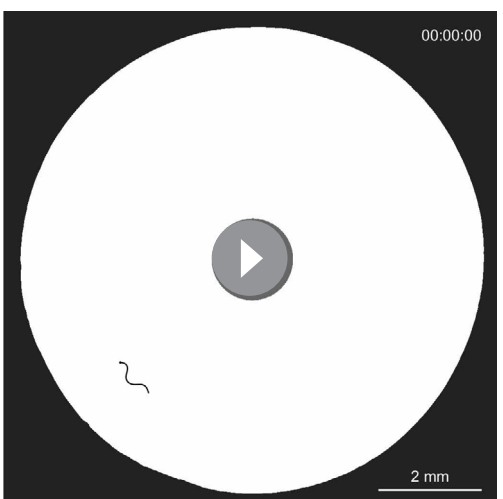

**Video 4.** Tracking the entire body of a foraging animal. The position of the entire body of an animal is shown as the animal forages for 1 hr in a 9-mm arena containing one small, low-density bacterial patch. The head is indicated by a small dot. The animal shown corresponds to the example animal observed in *Figure 3D*.

https://elifesciences.org/articles/103191/figures#video4

spent nearly 100% of time in the assay on bacterial patches, while at the lowest densities (0–0.3), animals spent chance levels of time on patch for the duration of the experiment. At intermediate densities (0.5–10) animals initially resided on patches at rates predicted by chance, later transitioning to more time spent on patch. Notably, this delayed increase in patch residence is density dependent with animals switching earlier in environments with patches of greater density.

Consistent with our previous observation that animals reside on patches for either short or long durations (*Figure 1E*), we observed two distinct patch encounter types: (1) short (less than 2 minute) patch visits and (2) long (often tens of minutes) patch visits (*Figure 2G*). We validated the existence of two patch types using Silverman's test (*Ahmed and Walther, 2012*; *Silverman, 1981*) for bimodality *Figure 2—figure supplement 6* and then, using a two-component GMM, classified all encounters based on the duration of these patch visits and animals' average velocity during the encounter (*Figure 2H*, *Figure 2—figure supplement 6*). We identified two behaviorally distinct clusters: (1) short-duration and fast velocity (*explore*) and (2) long-duration and slow velocity (*exploit*) encounters. The *explore* encounters suggest that an animal has *rejected* a patch, opting to continue exploration of the environment. In contrast, the *exploit* encounters suggest that an animal has *accepted* a patch and is consuming the bacteria within.

While we previously found that animals consistently slowed down upon encounter with the patch edge when foraging in relative density ~10 (*Figure 1J–M*), we observed that a large portion of short-duration, exploratory encounters were not accompanied by this slow down (*Figure 2—figure supplement 7A, B*), especially for the lowest bacterial densities tested. Therefore, to determine which exploratory encounters displayed evidence that the patch was detected by the animal, we further classified encounters by an animal's minimum velocity on patch, deceleration upon patch entry, and maximum change in the animal's velocity upon encounter (i.e., the difference between the peak velocity immediately before the patch encounter and the minimum velocity achieved during the patch encounter) (*Figure 2I*, *Figure 2—figure supplement 7*). We again observed two behaviorally distinct clusters: (1) large slowdown and slow on-patch velocity and (2) small slowdown and fast on-patch velocity. We validated the existence of these two clusters using Silverman's test (*Ahmed and Walther, 2012*; *Silverman, 1981*) for bimodality (*Figure 2—figure supplement 7*) and then classified these encounters as *sensing* or

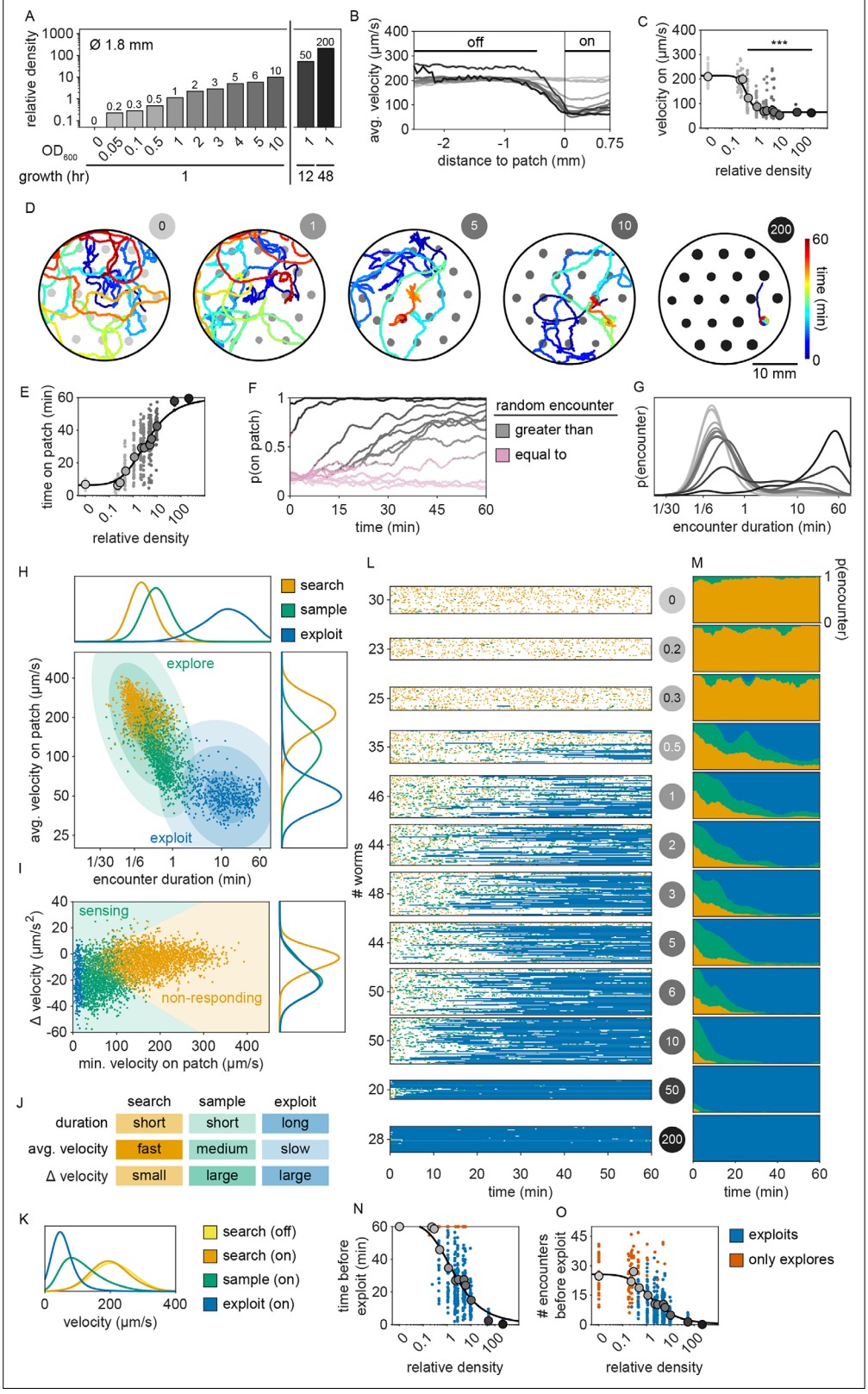

**Figure 2.** The timing of the switch from explore-to-exploit is density dependent. (**A**) The relative density (as estimated by fluorescently labeled OP50-GFP) is shown for small (~1.8 mm diameter) bacterial patches made by pipetting ~0.5 μl droplets of OP50 *E. coli* diluted in lysogeny broth (LB) to a range of optical densities OD₆₀₀ = {0, 0.05, 0.1, 0.5, 1, 2, 3, 4, 5, 10} and controlling growth time at room temperature (hours = {1, 12, 48}). For

*Figure 2 continued on next page*

*Figure 2 continued*

(**A–G**, **L–O**), gray-scale color saturation is proportional to the relative density of each condition and corresponds to labels in (**A**). (**B**) The mean velocities of animals foraging in environments containing patches matching one of the 12 bacterial densities are plotted as a function of the distance from the edge of bacterial patches (computed for 50 µm bins). (**C**) Animals' average on-patch velocity is plotted as a function of the relative density of bacteria. Compared to animals foraging among bacteria-free patches containing only LB (relative density 0), animals foraging on bacterial patches with relative density of 0.5 or greater display significantly slower on-patch velocities (one-tailed Mann–Whitney *U*-tests with Bonferroni correction). (**D**) The midbody location (colored to represent time in the experiment) of example animals foraging in environments containing patches (gray) of relative density 0, 1, 5, 10, and 200 is shown. (**E**) The total time each animal spent on patch is plotted as a function of the relative density of bacteria. Time on patch increased monotonically with increasing bacterial density (Kendall's *τ* correlation, p < 0.001) following a sigmoidal trend. (**F**) Smoothed median values of the probability of worms residing on patch over time for each density condition are plotted. Time points where observed probabilities of residing on patch either match (pink) or significantly exceed (gray) the probability of residing on patch if patch locations were semi-randomly permuted are indicated (one-tailed Fisher's exact test with Benjamini–Hochberg correction). (**G**) A kernel density estimate (KDE) of the distribution of encounter durations is plotted for each density condition. (**H**) For each encounter, the average velocity of the animal during the encounter and the duration of that encounter are plotted on a double-logarithmic plot with color representing the probabilities of clustering classification as *search* (orange), *sample* (green), or *exploit* (blue). Contours showing the first, second, and third standard deviation of the Gaussian mixture model (GMM) used to classify *explore* and *exploit* encounters are shown as shaded ellipses with saturation corresponding to standard deviation. KDEs for distributions of average on-patch velocity and encounter duration are plotted for each encounter type. (**I**) For each encounter, the minimum on-patch velocity and maximum change in velocity are plotted. Contours showing the separation of *sensing* and *non-responding* encounters as estimated by semi-supervised quadratic discriminant analysis (QDA) are indicated. A KDE for the distribution of the maximum change in velocity is plotted for each encounter type. (**J**) Features used to classify encounters as *search*, *sample*, or *exploit* are summarized. (**K**) KDEs of the distributions of animals' velocities are shown for all time points during search off and on patch as well as during sample and exploit encounters. (**L**) Ethograms of patch encounters (colored to represent the probability of classification as *search*, *sample*, and *exploit*) are shown for 443 individuals. (**M**) The average proportion of each encounter type over time is plotted. (**N**) Time elapsed and (**O**) number of encounters occurring prior to the first *exploitation* event are plotted for every animal (blue). In the event that no *exploitation* event occurred, the maximum observed time and encounters are plotted (red-orange). Both time and encounter number before *exploitation* decrease monotonically with increasing patch density (Kendall's *τ* correlation, p < 0.001) following a sigmoidal trend. Summary data for all animals (*N* = 443 total worms; *N* = 20–50 worms per condition) and encounters (*N* = 6560 total encounters; *N* = 46–876 encounters per condition) are shown in (**A–C**, **E–I**, **K–O**). Asterisks denote statistical significance (***p < 0.001). See also ***Figure 2—figure supplements 1–8*** and ***Videos 5 and 6***.

The online version of this article includes the following source data and figure supplement(s) for figure 2:

**Source data 1.** Relative density of bacterial patches in each condition in ***Figure 2A***.

**Source data 2.** Velocity of animals upon encounter with patch edge in ***Figure 2B***.

**Source data 3.** Velocity on patch in ***Figure 2C***.

**Source data 4.** Time on patch in ***Figure 2E***.

**Source data 5.** On-patch residence in ***Figure 2F***.

**Source data 6.** Duration of patch encounters in ***Figure 2G***.

**Source data 7.** Encounter classification as explore or exploit using a Gaussian mixture model (GMM) in ***Figure 2H***.

**Source data 8.** Encounter classification as sense or no response using semi-supervised quadratic discriminant analysis (QDA) in ***Figure 2I***.

**Source data 9.** Velocity during search, sample, and exploit behaviors in ***Figure 2K***.

**Source data 10.** Encounter classification as explore or exploit and sense or no response in ***Figure 2L***.

**Source data 11.** Probability of an encounter type over time in ***Figure 2M***.

**Source data 12.** Time before first exploitation in ***Figure 2N***.

**Source data 13.** Number of encounters before first exploitation in ***Figure 2O***.

**Figure supplement 1.** Obtaining intensity profiles for fluorescent bacterial patches.

**Figure supplement 2.** Example fluorescence profiles of bacterial patches under varied growth conditions.

**Figure supplement 3.** Quantifying the relative density of bacterial patches across conditions and time points.

*Figure 2 continued*

**Figure supplement 4.** Example traces of animals foraging in environments with varying bacterial density.

**Figure supplement 5.** Time- and density-dependent increase in patch residence.

**Figure supplement 6.** Classifying encounters as exploration or exploitation.

**Figure supplement 7.** Classifying encounters as sensing or non-responding.

**Figure supplement 8.** Classifying encounters as sensing or non-responding.

*non-responding*, respectively, using a semi-supervised quadratic discriminant analysis (QDA) approach (*Figure 2—figure supplement 8*, *Video 5*). The presence of a slowdown and the subsequent continuation of slower on-patch velocity suggest that an animal sensed the bacteria, while the absence of a slowdown response suggests that the animal either did not perceive the encounter or chose to ignore it. For simplicity, we define: (1) *non-responding* encounters as *searching* (i.e., on-patch behavior that matches off-patch behavior where an animal appears to be searching for food); (2) *sensing*, short-duration encounters as *sampling* (i.e., evaluating a patch's suitability as a food source, but ultimately *rejecting* the patch and continuing to explore the environment); and (3) *sensing*, long-duration encounters as *exploiting* (i.e., *accepting* a patch and consuming its bacteria) (*Figure 2J*). Consistent with our interpretation of *searching* encounters, the average velocities that animals achieved during *search* on (~200 μm/s) and off (~207 μm/s) patch were similar, especially when compared to markedly slower on-patch velocities during *sample* (~114 μm/s) and *exploit* (~57 μm/s) encounters (*Figure 2K*). In summary, we have identified three distinct patch encounter types: encounters lacking a slowdown that were either not detected or ignored (*search*) and encounters that were detected with the animal subsequently deciding to *reject* (*sample*) or *accept* (*exploit*) the patch.

We next characterized how the frequency of encounter types varied as a function of time and patch density. The behavior of 443 animals (20–50 worms per condition) was tracked and classified at each time point (*Figure 2L*, *Video 6*). In low-density conditions (0–0.3), animals spent nearly all of their time *searching* on or off patch. With increasing bacterial patch density, animals spent less time *searching* and more time *sampling* and *exploiting*. Animals foraging in medium-density patches (0.5–10) initially spent most of their time *sampling* patches upon encounter. However, animals eventually switched

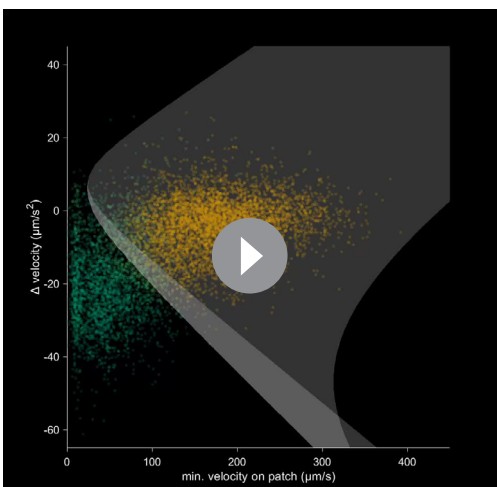

**Video 5.** Classifying encounters as *sensing* based on quantification of slowdown. Three metrics quantifying the magnitude of slowdown upon patch encounter were computed. A semi-self-supervised quadratic discriminant analysis was trained on a subset of labeled data and used to classify all encounters as *sensing* (green) or *non-responding* (orange). Data correspond with *Figure 2I*, *Figure 2—figure supplement 8A–C*.

https://elifesciences.org/articles/103191/figures#video5

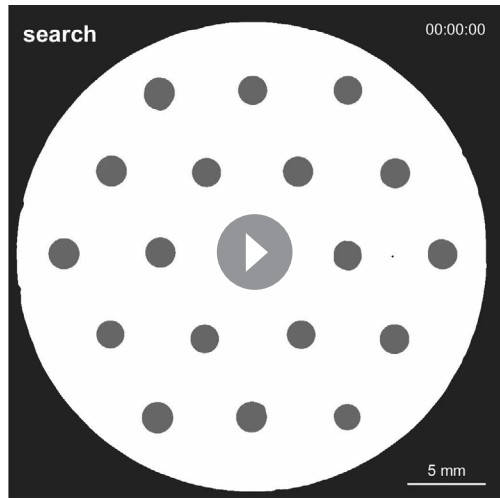

**Video 6.** Foraging behavior of an animal with encounters labeled as *search*, *sample*, or *exploit*. The midbody location of the animal in *Video 3* is shown and labeled by the probability of classification as *searching* (i.e., off-patch *exploration* and patch encounters lacking a slowdown response), *sampling* (i.e., patch encounters that were *sensed*, but not *exploited*), and *exploiting* at every time point. The animal shown corresponds to the example animal observed in *Figure 1A, B, H–K*.

https://elifesciences.org/articles/103191/figures#video6

to mostly *exploiting* patches. Animals foraging on high-density patches (50 and 200) almost exclusively *exploited*. Classification of on-patch behavior across all densities tested revealed a consistent switch from *exploratory* behaviors (*searching* and *sampling*) to *exploitation* (*Figure 2M*) with the timing of the switch occurring earlier for animals foraging in environments with higher bacterial patch density (*Figure 2N*). The delay in exploitation corresponded to a comparatively large number of initial exploratory encounters with animals on medium density conditions (0.5–10) *exploiting* after an average of 4.8–18.7 exploratory encounters (*Figure 2O*). To summarize, for densities similar to those experienced immediately preceding the experiment (50 and 200), animals immediately *exploit* available bacteria. On the other hand, for bacterial densities that appear to be below an animal's sensory threshold (0–0.3), animals never *exploit* patches and spend the entire 60-min assay *searching* for food. When foraging in medium-density patches (0.5–10), animals initially explore the environment via a combination of *searching* and *sampling* encounters and eventually switch to *exploiting*. Notably, the timing of this switch is density dependent with animals switching earlier when foraging on higher-density patches.

## Animals explore before exploiting even when only one patch is available

Given that bacterial patches within an environment were relatively invariable and that bacterial growth during the experiment was negligible (*Figure 2—figure supplement 3*), it was surprising that animals were often willing to *reject* 10+ patches before *exploiting* a patch of the same density as those previously *rejected* (*Figure 2O*). This *sampling* behavior may be beneficial when food is patchily distributed in the environment, as exploiting a dilute patch may result in a lost opportunity for future encounter with a higher quality patch. Thus, we hypothesized that an animal's willingness to *accept* a patch whose density matched that of a previously *rejected* patch could be a specific feature of a patchily distributed environment where numerous observations of individual patches could contribute to learning the features of a changing environment. To investigate this, we observed whether animals were willing to *reject* a patch numerous times before *accepting* it even when only one patch is available in the environment. We created environments with single large (~8.3 mm) or small (~1.8 mm) diameter patches with relative density ranging from 0 to ~400 (*Figure 3A–C*). Behavior was tracked for 60 min as animals foraged in these environments (*Figure 3D, E*, *Figure 3—figure supplements 1 and 2*). We found that even in these single patch environments, animals initially *explore* (*search* and *sample*) before *exploiting* (*Figure 3F, G*) in a density-dependent manner. Just as when foraging in a patchily distributed environment (*Figure 2N*), animals *explored* (*search* and *sample*) the small and large single patches numerous times prior to *exploiting*, with the timing of this switch occurring significantly earlier with increasing bacterial density (*Figure 3H, I*). These results suggest that the explore-then-exploit strategy employed by *C. elegans* is not specific to a patchily distributed environment. Rather, animals may *sample* the same patch numerous times before deciding to *exploit*, especially when foraging on dilute patches. The decision of when to stop *exploring* and start *exploiting* is highly dependent upon the density of bacteria within a patch, with the switch occurring earlier when animals encounter patches of higher density.

## External and internal sensory information guide the decision to explore or exploit

While our behavioral analysis demonstrated that *C. elegans* modulate their decision to *explore* or *exploit* in a density-dependent manner, it remains unclear whether these animals use more complex cognitive processes such as learning and memory to guide their foraging decision. It is plausible that animals are making decisions using simple heuristics given only available sensory information about the current patch. However, previous studies have shown that food-related behaviors in *C. elegans* can be driven by internal states (*Scheer and Bargmann, 2023*) as well as memories of the environment (*Calhoun et al., 2015*; *Shtonda and Avery, 2006*; *Pradhan et al., 2019*). Thus, to better understand the factors driving the exploitation decision, we implemented a generalized linear model (GLM) to test how foraging decisions are influenced by the density of the current patch as well as additional food-related factors such as an animal's level of satiety and prior experience (*Figure 4A*).

We considered that the probability of exploiting a patch upon any given encounter can be described by a logistic function:

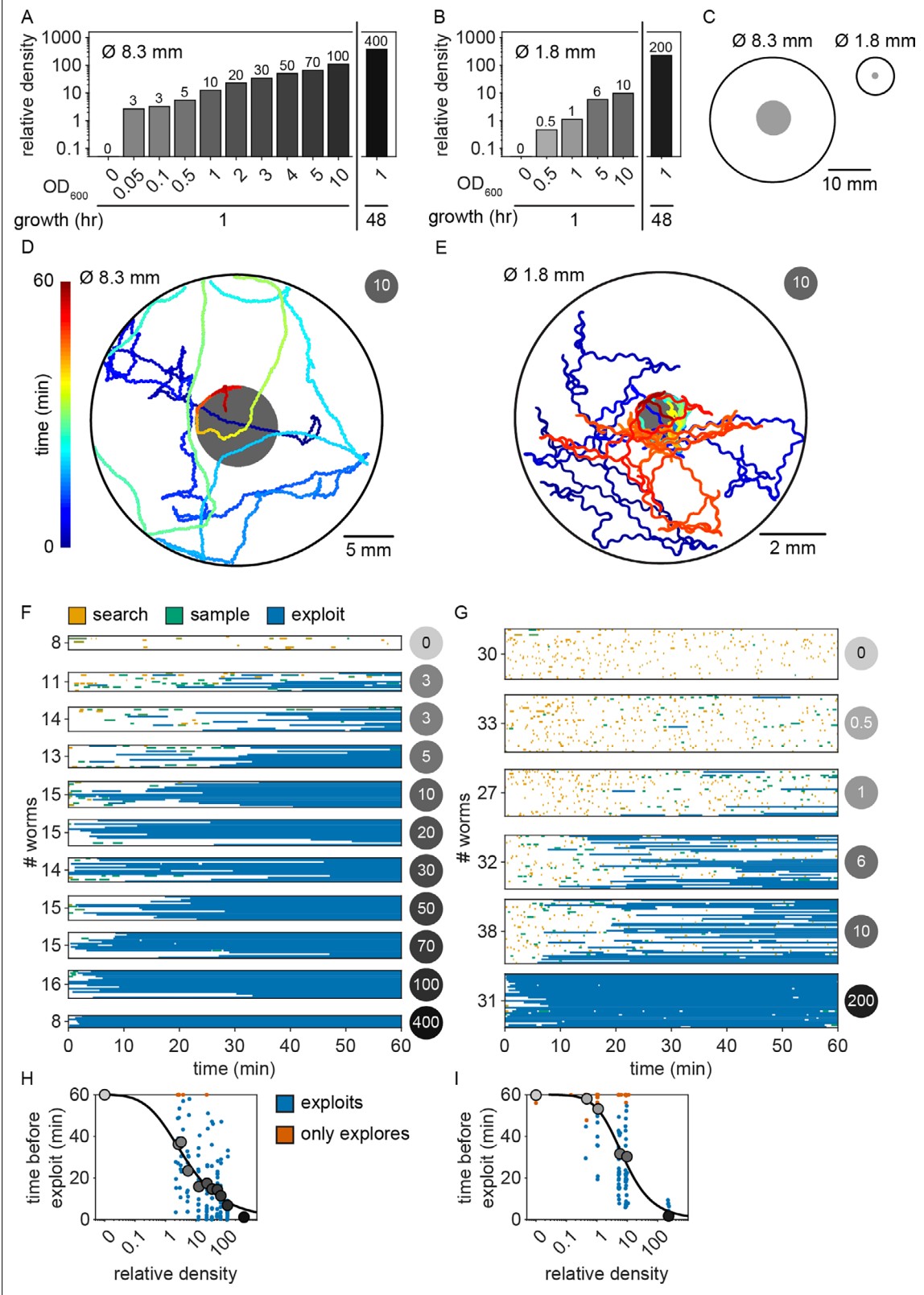

**Figure 3.** *C. elegans* explore first even when only one patch is available. (**A**) Relative density of large (~8.3 mm diameter) and (**B**) small (~1.8 mm diameter) bacterial patches was varied by pipetting 20 and 0.5 µl, respectively, droplets of OP50 *E. coli* diluted in lysogeny broth (LB) to a range of optical densities $OD_{600}$ = {0, 0.05, 0.1, 0.5, 1, 2, 3, 4, 5, 10} and controlling growth time at room temperature (hours = {1, 48}). (**C**) Large patches were formed in the center of 30-mm arenas while small patches were formed in the center of 9-mm arenas. The midbody location (colored to represent time

*Figure 3 continued on next page*

*Figure 3 continued*

in the experiment) of example animals foraging in environments containing a single (**D**) large or (**E**) small patch (gray) is shown. Ethograms of patch encounters (colored to represent the probability of classification as *search*, *sample*, and *exploit*) for (**F**) 144 individuals (8–15 per condition) foraging on a single large patch and (**G**) 191 individuals (27–38 per condition) foraging on a single small patch are shown. Time elapsed prior to the first *exploitation* event (blue) for animals foraging on (**H**) large and (**I**) small patches is plotted for every animal. When *exploitation* was not observed, time elapsed in the experiment is plotted (red-orange). Animals in large and small patch environments *exploited* higher-density patches significantly earlier (Kendall's $\tau$ correlation, p < 0.001) following a sigmoidal trend. See also *Figure 3—figure supplements 1 and 2*.

The online version of this article includes the following source data and figure supplement(s) for figure 3:

**Source data 1.** Relative density of large (20 µl) bacterial patches in *Figure 3A*.

**Source data 2.** Relative density of small (0.5 µl) bacterial patches in *Figure 3B*.

**Source data 3.** Encounter classification as explore or exploit and sense or no response in *Figure 3F*.

**Source data 4.** Encounter classification as explore or exploit and sense or no response in *Figure 3G*.

**Source data 5.** Time before first exploitation in *Figure 3H*.

**Source data 6.** Time before first exploitation in *Figure 3I*.

**Figure supplement 1.** Example traces of animals foraging in environments with one large bacterial patch.

**Figure supplement 2.** Example traces of animals foraging in environments with one small bacterial patch.

$$p\left(y_k = 1|\boldsymbol{\beta} \cdot \boldsymbol{x}_k\right) \;=\; \frac{1}{1 + e^{-\boldsymbol{\beta} \cdot \boldsymbol{x}_k}},$$

where $p\left(y_k = 1|\boldsymbol{\beta} \cdot \boldsymbol{x}_k\right)$ represents the conditional probability that an animal exploited during patch encounter $k$; $\boldsymbol{x}_k \in R^n$ is a vector of covariates; and $\boldsymbol{\beta} \in R^n$ is a vector of weights that describes how much each covariate influences the animal's choice. In models compared here, $\boldsymbol{x}_k$ includes a combination of a constant element and covariates that may empirically influence the decision to *exploit*. We considered the simplest model where every encounter is independent with fixed probability of *exploiting* (i.e., $p\left(y_k = 1|\beta_0\right)$) where $\beta_0$ represents the average animal's propensity to *exploit*. We compared this null model against a set of nested models containing covariates that vary from encounter to encounter and relate to: (1) the relative density of the encountered patch $k$ ($\rho_k$), (2) the duration of time spent off food since departing the last *exploited* patch ($\tau_s$), (3) the relative density of the patch encountered immediately before encounter $k$ ($\rho_h$), and (4) the relative density of the last *exploited* patch ($\rho_e$). Using this model design (i.e., $\boldsymbol{\beta} \cdot \boldsymbol{x}_k = \beta_0 + \beta_k\rho_k + \beta_s\tau_s + \beta_h\rho_h + \beta_e\rho_e$), we assessed how each of these covariates influences the probability of *exploitation* at each patch encounter $k$ (*Figure 4A*).

Given that our aim is to understand what factors influence an animal's decision to *explore* or *exploit*, we rationalized that encounters where the animal did not detect the presence of bacteria were unlikely to contribute to decision-making. Therefore, we excluded *searching* encounters from our analysis where animals did *not respond* to the bacterial patch. To account for the uncertainty in our classification of *sensing*, we probabilistically included patch encounters with frequency equal to the probability that the bacterial patch was *sensed* (*Figure 4B*) as previously estimated by semi-supervised QDA (*Figure 2I*, *Figure 2—figure supplement 8*). In doing so, we simulated sets of observations of *sensed* (*sample* and *exploit*) encounters. Further, to account for the uncertainty in our classification of patch encounters as *exploration* or *exploitation*, we fit our GLM to the probability of *exploiting* $p\left(y_k = 1|z_k\right)$ as previously estimated by GMM (*Figure 2H*, *Figure 2—figure supplement 6*) rather than fitting to direct observations of exploitation $y_k$ (see Models of exploitation probability).

When selecting covariates for our model, we calculated the Bayesian information criterion – a model selection test to avoid overfitting by penalizing increases in the number of parameters – and observed the best model performance when all covariates were included (*Figure 4—figure supplement 1A–B*). Further, we found that all covariates (i.e., $\rho_k$, $\tau_s$, $\rho_h$, and $\rho_e$) significantly contribute to the decision to *exploit* $p\left(y_k = 1|\boldsymbol{\beta} \cdot \boldsymbol{x}_k\right)$ as indicated by coefficient values significantly greater than or less than zero (*Figure 4C*, *Figure 4—figure supplement 1*). Specifically, we found that $\beta_k$ and $\beta_s$ are significantly greater than zero, which suggests that animals are more likely to *exploit* patches with increasing patch density $\rho_k$ and duration of food deprivation $\tau_s$. On the other hand, $\beta_h$ and $\beta_e$ are significantly negative, which suggests that the greater the density of a recently encountered or *exploited* patch, $\rho_h$ and $\rho_e$, the less likely an animal should be willing to *exploit* the current patch.

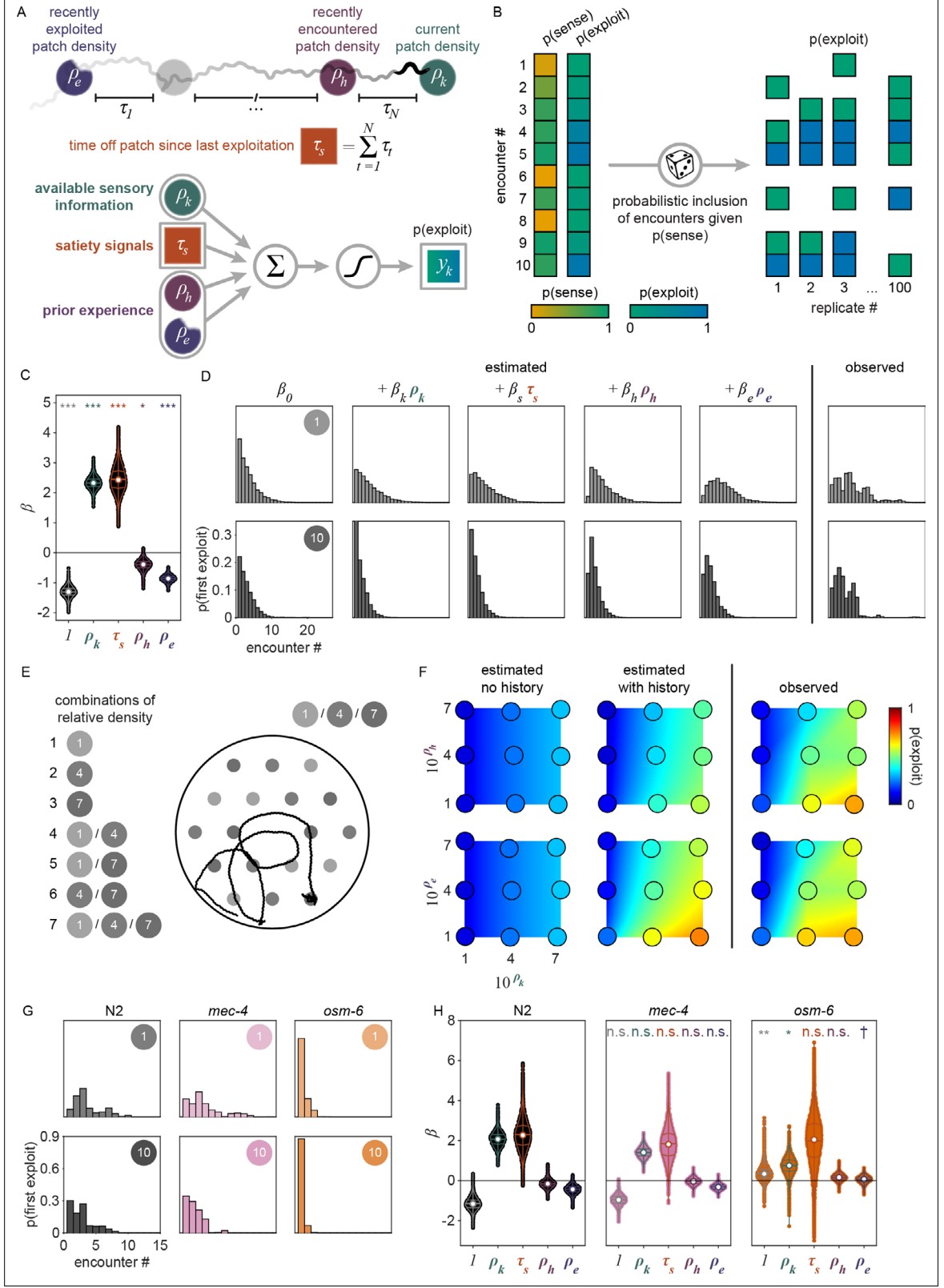

**Figure 4.** Exploitation decisions are driven by available sensory information, satiety, and prior experience. (**A**) Schematic of the covariates $x_k$ used in our logistic regression model to predict the probability of *exploiting* a patch upon any given encounter $p\left(y_k = 1|\boldsymbol{\beta} \cdot \boldsymbol{x}_k\right)$. $\boldsymbol{x}_k$ includes the relative density of the encountered patch $k$ ($\rho_k$), the duration of time spent off food since departing the last *exploited* patch ($\tau_s$), the relative density of the patch encountered immediately before encounter $k$ ($\rho_k$), and the relative density of the last *exploited* patch ($\rho_e$). (**B**) To account for uncertainty in

*Figure 4 continued on next page*

*Figure 4 continued*

our classification of sensation, we produced 100 sets of observations wherein we probabilistically included *sensed* encounters ($v_k = 1$) estimated from the distribution $v_k | w_k \sim \text{Bern}\left(p\left(v_k = 1|w_k\right)\right)$ where $p\left(v_k = 1|w_k\right)$ is the probability that the patch was *sensed* as estimated in *Figure 2I*. To account for uncertainty in our classification of *exploitation*, we substitute the response variable $y_k$ with our estimate of the probability that the patch was *exploited* $p\left(y_k = 1|z_k\right)$ as estimated in *Figure 2H*, a procedure analogous to including *exploitations* ($y_k = 1$) estimated from the distribution $y_k | z_k \sim \text{Bern}\left(p\left(y_k = 1|z_k\right)\right)$. A schematic of these procedures is shown (see Models of exploitation probability). (**C**) Coefficient values for each covariate in the GLM were estimated across 50,000 replicates (500 replicates of hierarchically bootstrapped animals in combination with 100 sets of probabilistically *sensed* encounters). All coefficients are significantly greater than or less than 0 (two-tailed, one-sample bootstrap hypothesis tests with Bonferroni correction). (**D**) Using observed $p\left(y_k = 1|z_k\right)$ and estimated $p\left(y_k = 1|\boldsymbol{\beta} \cdot \boldsymbol{x}_k\right)$ probabilities of *exploitation*, *exploitation* events were simulated from a Bernoulli distribution. Distributions of the probability of first *exploiting* as a function of the number of encounters as estimated by the model with covariates added one-by-one and as observed are shown for animals foraging in single-density, multi-patch environments of relative density 1 or 10. (**E**) A schematic of multi-density, multi-patch assays where animals foraged in environments containing small (~1.8 mm diameter) bacterial patches (gray) of varying combinations of OP50 *E. coli* with relative densities 1, 4, and 7 are shown. (**F**) The probability of exploitation as estimated with ($p\left(y_k = 1|\beta_0 + \beta_k\rho_k + \beta_s\tau_s + \beta_h\rho_h + \beta_e\rho_e\right)$) and without ($p\left(y_k = 1|\beta_0 + \beta_k\rho_k + \beta_s\tau_s\right)$) the history-dependent terms and as observed $p\left(y_k = 1|z_k\right)$ is shown for every pairing of current patch density ($\rho_k$) combined with the density of recently encountered ($\rho_h$) or *exploited* ($\rho_e$) patches. Heat maps for observed values were interpolated between the nine patch density pairings tested. (**G**) *Exploitation* events were simulated from the distribution $y_k | z_k \sim \text{Bern}\left(p\left(y_k = 1|z_k\right)\right)$ and used to calculate the probability of observing *exploitation* for the first time as a function of the number of encounters for N2, *mec-4*, and *osm-6* animals foraging in single-density, multi-patch environments of relative density 1 or 10. (**H**) Coefficient values for each covariate were estimated using ridge regression models for each strain across 50,000 replicates. A subset of these coefficients significantly varied between wild-type and mutant strains (mean of differences tests with Benjamini–Hochberg correction). Summary data for all animals ($N$ = 443 total worms; $N$ = 20–50 worms per condition) and encounters ($\mu$ = 2659.8, $\sigma$ = 15.9) in the single-density, multi-patch assay are shown in (**C**, **D**). Summary data for all animals ($N$ = 198 total worms; $N$ = 20–40 worms per condition) and *sensed* encounters ($\mu$ = 1724.6, $\sigma$ = 9.9) in the multi-density, multi-patch assay are shown in (**F**). Summary data for all wild-type and mutant animals ($N$ = 221 total worms; $N$ = 14–44 worms per condition) and encounters ($N$ = 1352 total encounters) in a single-density, multi-patch assay are shown in (**G**, **H**). Violin plots in (**C**, **H**) show the kernel density estimate (KDE) and quartiles for each measure. Asterisks denote statistical significance ($^\dagger p_{\text{unadjusted}}$ < 0.05; *p < 0.05; **p < 0.01; ***p < 0.001). See also *Figure 4—figure supplements 1–5*.

The online version of this article includes the following source data and figure supplement(s) for figure 4:

**Source data 1.** Coefficients of linear regression model in *Figure 4C*.

**Source data 2.** Probability of exploitation occurring for the first time as a function of the number of encounters in *Figure 4D*.

**Source data 3.** Observed and predicted (with and without history dependence) posterior probabilities of exploitation in *Figure 4F*.

**Source data 4.** Probability of exploitation occurring for the first time as a function of the number of encounters for animals with sensory mutations in *Figure 4G*.

**Source data 5.** Coefficients of linear regression model in *Figure 4H*.

**Figure supplement 1.** Model selection for nested GLMs.

**Figure supplement 2.** Probability of first exploitation as a function of the number of encounters.

**Figure supplement 3.** Exploitation decision of food-deprived animals is well predicted by the model.

**Figure supplement 4.** Example traces and classified behavior of animals foraging in environments with multiple patch densities.

**Figure supplement 5.** Example traces and classified behavior of animals with chemosensory or mechanosensory deficiencies.

In order to better understand how each of the covariates in our model affects the *exploitation* decision, we examined the probability of observing *exploitation* for the first time as a function of the cumulative number of patches an animal encountered. Exploitation events were simulated from a Bernoulli distribution with probability estimated by the model (i.e., $y_k | x_k \sim \text{Bern}\left(p\left(y_k = 1|\boldsymbol{\beta} \cdot \boldsymbol{x}_k\right)\right)$) or observed from our classification of *exploitation* (i.e., $y_k | z_k \sim \text{Bern}\left(p\left(y_k = 1|z_k\right)\right)$). Although the probability of *exploiting* for the first time was not expressly fit by our model, this derived metric highlights the model's ability to predict the delay in *exploitation* observed for animals foraging on low to medium density bacterial patches in our experiments (*Figure 4D*, *Figure 4—figure supplement 2*). For the simplest model where every encounter is independent with fixed probability of *exploiting* $p\left(y_k = 1|\beta_0\right)$, we find that the probability of *exploiting* for the first time is greatest for the first encounter and decreases for every subsequent encounter (*Figure 4D*, *Figure 4—figure supplement 2*), fitting a geometric distribution.

We next considered whether the observed delay in *exploitation* could be explained by a simple density-dependent model (i.e., $p\left(y_k = 1|\beta_0 + \beta_k\rho_k\right)$). This model predicts that animals are more likely to *exploit* bacterial patches of greater density with probability proportional to the relative density of the current patch $\rho_k$. While this density-dependent model accurately predicted that the first *exploitation* occurs earlier on average for higher density patches, it also predicted that, regardless of patch density, the first *exploitation* is most frequently observed on the first encounter, which did not match

the observed distribution (*Figure 4D*, *Figure 4—figure supplement 2*). We subsequently considered that the onset of food deprivation during *exploration* of these low-density environments could drive the observed time-dependent delay in *exploitation*. We assume that animals become satiated during *exploitation* and use the duration of time an animal spent searching off-food since the last *exploitation* event (i.e., $\tau_s$) as a proxy for the animal's likely decrease in satiety. With the addition of this satiety-related signal (i.e., $p\left(y_k = 1|\beta_0 + \beta_k\rho_k + \beta_s\tau_s\right)$), we observed a slight delay in when animals are predicted to first *exploit* for the lowest density conditions tested (*Figure 4D*, *Figure 4—figure supplement 2*). To validate our finding that animals use a satiety-related signal to inform decision-making, we applied our model to observations of animals foraging in patchily distributed environments following 3 hr of food deprivation (*Figure 4—figure supplement 3A–B*). We found that only with the inclusion of a satiety-related covariate in our model were we able to reliably predict the immediate *exploitation* that food-deprived animals exhibit (*Figure 4—figure supplement 3C*).

We next considered the alternative possibility that this time-dependent signal could actually be related to the physical stress induced by the process of moving animals into the experimental arena rather than the onset of food deprivation. While we used an agar plug method to mitigate the stress of moving (see Nematode cultures), we sought to examine the impact of satiety as compared to transfer-induced stress on the *exploitation* decision. We quantified both $\tau_s$, the duration of time an animal spent searching off-food since the last *exploitation* event, and $\tau_t$, the duration of time since transfer (i.e., the time elapsed in the experiment). However, as our initial experiments were only 1 hr, we typically observed only one *exploitation* event. Resultantly, $\tau_s$ and $\tau_t$ were highly correlated, and their individual contributions to the model could not be easily parsed. Therefore, we ran additional experiments with animals foraging in environments with multiple patch densities over 2 hr (*Figure 4E*, *Figure 4—figure supplement 4A–C*), which led to observations containing multiple *exploitation* events and a resultant decorrelation of $\tau_s$ and $\tau_t$. When the model was re-fit to this 2-hr data set using both $\tau_s$ and $\tau_t$ covariates (i.e., $p\left(y_k = 1|\beta_0 + \beta_k\rho_k + \beta_s\tau_s + \beta_t\tau_t + \beta_h\rho_h + \beta_e\rho_e\right)$), only the satiety-related covariate $\tau_s$ was statistically significant (*Figure 4—figure supplement 4D*). This result suggests that the time-dependent change in *exploitation* is likely due to a satiety-related signal rather than a stress-related signal. Altogether, these results suggest that animals use available external and internal sensory information to guide the decision to *exploit*.

## Prior experience guides the decision to *explore* or *exploit*

Although the current patch density and satiety covariates significantly influence the decision to *exploit*, this model (i.e., $p\left(y_k = 1|\beta_0 + \beta_k\rho_k + \beta_s\tau_s\right)$) does not accurately predict the extent that *exploitation* was delayed. We hypothesized that because animals had experienced very densely seeded bacterial patches immediately prior to the assay, animals may have built an expectation that high-density bacterial patches are available in the new environment. We would therefore expect an initial suppression in the probability of *exploiting* while animals updated their expectation using information learned from *sampling* patches in the current environment. To test whether the decision to *exploit* could be influenced by prior experience, we added a history-dependent covariate to our model (i.e., $p\left(y_k = 1|\beta_0 + \beta_k\rho_k + \beta_s\tau_s + \beta_h\rho_h\right)$ where $\rho_h$ is the relative density of the patch encountered immediately before encounter $k$). We find that inclusion of the density of the most recently encountered patch significantly improves our prediction of *exploitation* (*Figure 4D*, *Figure 4—figure supplement 2*). We then considered whether the animal's expectation of bacterial density in the environment might be a longer lasting memory. Specifically, we considered if the relative density of the most recently *exploited* patch $\rho_e$ could have influenced the animal's decision to *exploit*. The addition of this second history-dependent term to our model (i.e., $p\left(y_k = 1|\beta_0 + \beta_k\rho_k + \beta_s\tau_s + \beta_h\rho_h + \beta_e\rho_e\right)$) further improved the fit of our model (*Figure 4D*, *Figure 4—figure supplement 2*), suggesting that *C. elegans* use recent experiences of encountered and *exploited* patches to guide their decision to *exploit*. Animals may use this learned information to create and update an expectation for the relative density of bacterial patches available in their environment and compare that estimate against the density of a subsequently encountered patch. Altogether, these results suggest that animals use recent experiences to estimate available resources and modulate their decision to *explore* or *exploit*, a strategy that is highly beneficial for behavioral adaptation in a changing environment.

To validate the role of prior experience in guiding foraging decisions, we observed animals foraging in environments with multiple patch densities (*Figure 4E*, *Figure 4—figure supplement 4A–C*). Our

model predicts that the lower the density of recently encountered and/or *exploited* patches, the more likely an animal should be to *exploit* a subsequent patch. Without the history-dependent terms $\rho_h$ and $\rho_e$, our model predicts that no relationship should exist between prior experience and the probability of *exploiting* the current patch. To test which model best fit the animals' behavior, we calculated the conditional probabilities of *exploiting* $p\left(y_k = 1 | \boldsymbol{\beta} \cdot \boldsymbol{x}_k\right)$ using the coefficients $\beta$ learned previously (*Figure 4C*) for this new multi-density, multi-patch data set containing more nuanced combinations of current ($\rho_k$) and recent patch ($\rho_h$ and $\rho_e$) density. We observed significantly improved predictions of *exploitation* probability for the model including prior experience as compared to the model without the terms $\rho_h$ and $\rho_e$ (*Figure 4F*, *Figure 4—figure supplement 4E*) as confirmed by a likelihood-ratio test. We observed an augmentation in *exploitation* when animals move from low to high density (e.g., 1–4 or 1–7) as predicted by the model with history dependence, but not the model without. It is important to note that this improvement in prediction was achieved without re-fitting the model to this new multi-density data set. In summary, our modeling results suggest that animals evaluate bacterial density, monitor internal signals of satiety, and leverage recent experiences to drive their decision to *exploit* a bacterial patch upon encounter.

## Ciliated sensory neurons are required for the evaluation of patch density

The quantitative modeling applied in this study provides a useful framework for testing the influence of behaviorally relevant features on decision-making. To validate this approach and further elucidate how animals evaluate current and recent patch density, we assessed the foraging behavior of mutants with sensory deficiencies (*Figure 4—figure supplement 5*). We found that animals with reduced function of non-ciliated mechanoreceptor neurons due to a null mutation in the *mec-4* gene behave similarly to wild-type N2 animals, displaying increasingly delayed *exploitation* as patch density decreases (*Figure 4G*, *Figure 4—figure supplement 5*). Contrastingly, animals with reduced function of ciliated chemoreceptor and mechanoreceptor neurons due to a null mutation in the *osm-6* gene displayed immediate *exploitation* even while foraging on low-density patches (*Figure 4G*, *Figure 4—figure supplement 5*). These results indicate altered decision-making in the *osm-6* mutants. Specifically, it appears that the influence of patch density on the *exploitation* decision is reduced in *osm-6* mutants and that these animals are more likely to *accept* a patch by default.

To further elaborate on these findings, we fit our GLM to each of the three data sets (i.e., N2, *mec-4*, and *osm-6*) and assessed the difference in the estimated distributions of coefficients $\beta$ between mutant and wild-type models (*Figure 4H*). We hypothesized that the density-related covariates ($\rho_k$, $\rho_h$, and $\rho_e$) would have smaller magnitude (i.e., less influence on decision-making) for animals with reduced sensation of bacterial density. Consistent with this hypothesis, we found a significant reduction in the magnitude of the influence of current patch density (i.e., $\beta_k$) on the decision to exploit in *osm-6* mutants. Further, we found a significant increase in the intercept $\beta_0$ in *osm-6* mutants which corresponds to the overall increase in *exploitation* observed in these animals (*Figure 4G*, *Figure 4—figure supplement 5*). These results suggest that the ciliated sensory neurons play an important role in evaluating the density of the current patch and that when sensation is impaired, animals modify their default behavioral strategy to prioritize *exploitation*. We also observed a marginal increase in our estimate of the influence of recently *exploited* patches (i.e., $\beta_e$) in *osm-6* mutants, which suggests that reduced efficacy in the evaluation of current patch density also affects learning and memory of patch density. No significant changes were detected in the *mec-4* mutants. Although we cannot rule out a role for non-ciliated mechanoreceptor neurons in contributing to the assessment of patch density, this modality likely provides less salient information. Altogether, these results suggest that ciliated sensory neurons are necessary for the evaluation of patch density and that when the function of these neurons is impaired, animals prioritize *exploitation*. Further, these results demonstrate that this quantitative modeling approach can be used to probe mechanisms underlying decision-making.

## Discussion

Like all foraging animals, *C. elegans* must choose between *exploiting* an environment for known resources and *exploring* it for potentially better opportunities elsewhere (*Stephens, 2008*; *Schoener, 1971*; *Pyke et al., 1977*; *Stephens and Krebs, 1986*; *Stephens et al., 2007*; *Nonacs, 2001*). Previous

studies have shown that *C. elegans* use *stay–switch* decisions to efficiently explore a patchily distributed environment and exploit the bacteria within (*Iwanir et al., 2016*; *Pradhan et al., 2019*; *Gloria-Soria and Azevedo, 2008*; *Madirolas et al., 2023*). In this study, we show that *C. elegans* also make *accept–reject* decisions while foraging. Specifically, we observe that animals often initially *reject* low-density bacterial patches, opting to prioritize exploration of the environment before switching to a more exploitatory foraging strategy during subsequent encounters. This explore-then-exploit strategy was consistent across a range of bacterial patch densities, sizes, and distributions. In order to better understand this phenomenon and identify the factors that contribute to the decision to *exploit*, we leveraged a quantitative modeling approach to hypothesis testing. We found that animals do not employ a default explore-then-exploit strategy; rather, this behavior reflects a series of *accept–reject* decisions guided by multimodal information related to an animal's environment and internal state. Specifically, we show that each decision to *explore* or *exploit* is informed by available sensory information, internal satiety signals, and learned environmental statistics related to the bacterial density of recently encountered and *exploited* patches.

## *C. elegans* make *accept–reject* foraging decisions

While *C. elegans* have been shown to alter food preferences in a diet choice assay, this behavior has only been described by a set of *stay–switch* decisions (*Scheer and Bargmann, 2023*; *Shtonda and Avery, 2006*; *Madirolas et al., 2023*). Specifically, *C. elegans* were found to modulate their patch-leaving frequency to *stay* on high-quality bacterial patches and *switch* with declining food quality and quantity (*Bendesky et al., 2011*; *Shtonda and Avery, 2006*; *Milward et al., 2011*; *Olofsson, 2014*). The overall effect of leaving low-quality patches more frequently is that animals are more likely to reside on high-quality patches. Thus, *stay–switch* decision-making enables *C. elegans* to allocate time spent between exploring or exploiting in a manner consistent with optimal foraging theory (*Stephens, 2008*; *Schoener, 1971*; *Pyke et al., 1977*; *Stephens and Krebs, 1986*; *Stephens et al., 2007*; *Nonacs, 2001*). However, optimal foraging theory also predicts that an animal should never exploit (i.e., always *reject*) a low-density patch if higher-density patches are sufficiently abundant, as inclusion of the low-density patch type in the animals' diet decreases the overall rate of energy gained (*Stephens, 2008*; *Stephens and Krebs, 1986*). Thus, while leaving low-density patches more frequently drives an overall increase in the rate of energy gained, it can still result in sub-optimal foraging behavior.

Here, we demonstrate that *C. elegans* are capable of making *accept–reject* decisions upon encounter with bacterial patches. We employed quantitative methods – GMM and semi-supervised QDA – to estimate whether animals *sensed* a patch and, if they did, whether they *accepted* or *rejected* it. We found that when transferred from high- to low-density bacterial patches, animals initially *rejected* several patches, opting to continue exploration of the environment, and that this *reject* decision did not reflect an animal's inability to detect the presence of bacteria in these dilute patches. Further, we found that the decision to *explore* or *exploit* is density-dependent and robust across a range of bacterial densities, distributions, and sizes. The initial preference for *exploration* during foraging on low-density patches suggests that animals may be *searching* for denser bacterial patches such as those experienced immediately preceding the assay and throughout development. This explore-then-exploit strategy is advantageous for animals when bacterial patch density is low or variable. On the other hand, we observed immediate *exploitation* of high-density patches. In these environments, an explore-then-exploit strategy is disadvantageous, as delaying *exploitation* when patch density is already high results in spending more energy on food search with no expected increase in energy gained during *exploitation* of a subsequently encountered patch. Thus, consistent with an energy-maximizing optimal forager, *C. elegans* initially *explores* when bacterial patch density is low but immediately *exploits* when bacterial patch density is sufficiently high. Altogether, our results demonstrate that *C. elegans* make a distinct decision to either *explore* or *exploit* a patch upon encounter and that the density dependence of this decision is consistent with predictions from optimal foraging theory.

The concepts of *accept–reject* and *stay–switch* decision-making are well described in foraging theory literature (see *Pyke et al., 1977* for a review on the subject) and represent the questions 'Do you want to eat it?' and, if so, 'How long do you want to eat it for?'. These concepts are easily distinguishable for animals that forage using the following framework: (1) search for prey, (2) encounter prey from a distance, (3) identify prey type, (4) decide to pursue (*accept–reject* decision), (5) pursue

and capture the prey, (6) exploit prey, and (7) decide to stop exploiting and start searching again (*stay–switch* decision). However, in some scenarios, animals must physically encounter prey prior to identification. In these cases where pursuit and capture are not visualized, it is harder to distinguish between *accept–reject* and *stay–switch* decisions. In our experiments, *C. elegans* do not appear to detect the presence of bacterial patches prior to encounter as slowdown only occurs within one body length of the patch edge (*Figure 2B*). Further, we find significant bimodality in encounter duration (*Figure 2H*) where short-duration (*exploratory*) encounters appear to represent a lower bound where animals spend the minimum amount of time possible on a patch (less than 2 min), which we interpret as a *rejection* of the patch. On the other hand, *exploitatory* encounters span a large range of durations from 2 to 60+ min which we interpret as an initial *acceptance* of the patch followed by a series of *stay–switch* decisions which determine the overall duration of the encounter. While we could certainly model our data using only *stay–switch* decision-making, we ascertain that an encounter of minimal duration is better interpreted ethologically as a *rejection* rather than as an immediate *switch* decision. In accordance with this interpretation, we hypothesize that if *C. elegans* were able to detect the presence of a bacterial patch prior to encountering it, it is possible that animals may be observed to navigate toward (*accept*) or away from (*reject*) a patch. However, it could be the case that, for *C. elegans*, *sampling* the patch is prerequisite to an *accept-reject* decision which would suggest that gustation, mechanosensation, and/or post-ingestive feedback may guide *C. elegans* foraging decisions.

Our finding that *C. elegans* are willing to *reject* numerous encountered patches is somewhat surprising, especially given that *C. elegans* likely possess limited spatial memory (*Calhoun et al., 2015*) and are likely unaware of how many patches are in the environment as evidenced by their willingness to *reject* a patch even when only one patch is available (*Figure 3*). Taken together, this suggests that animals are 'knowingly' passing up an opportunity to *exploit* an immediately available food source for the potential opportunity to *exploit* preferred food later. This type of behavior has been described in humans and other animals in the context of delayed gratification (*Fawcett et al., 2012*; *Susini et al., 2021*), intertemporal choice (*Fawcett et al., 2012*; *Constantino and Daw, 2015*; *Carter et al., 2015*), self-control (*Hayden, 2019*; *Stephens and Anderson, 2001*), and optimal stopping (*Ferguson, 1967*; *Robbins, 1970*). Notably, few animals display the capacity for delayed gratification when given the option between eating an immediately available but less preferred food item and waiting for a preferred food item (*Mischel and Ebbesen, 1970*; *Rosati et al., 2007*; *Schnell et al., 2021*; *Range et al., 2020*). Even fewer animals are willing to wait if the delayed reward is not visible or increases in quantity rather than quality (*Hillemann et al., 2014*; *Miller et al., 2020*). Our observation that *C. elegans* can exhibit delayed gratification suggests that this and similar processes require less complex cognition than previously considered or that we may have underestimated the cognitive capacity of *C. elegans*.

## External and internal sensory information guide foraging decision-making

*Stay–switch* decisions in *C. elegans* are strongly modulated by a variety of internal and external sensory cues over a range of behavioral timescales. For example, patch-leaving frequency can be modulated by an animal's prior experience (*Calhoun et al., 2015*; *Pradhan et al., 2019*), metabolic status, arousal state (*Scheer and Bargmann, 2023*), as well as the presence of pathogens (*Pujol et al., 2001*; *Melo and Ruvkun, 2012*), chemorepellents (*Pradel et al., 2007*), predators (*Quach and Chalasani, 2022*; *Pribadi et al., 2023*), and varying levels of environmental $O_2$ and $CO_2$ (*Milward et al., 2011*; *Busch and Olofsson, 2012*).

Here, we used a quantitative framework to show that recent experiences (i.e., the density of current and recently encountered and *exploited* bacterial patches) guide *accept–reject* decision-making. In our study and others (*Madirolas et al., 2023*; *Sawin et al., 2000*), *C. elegans* appear to be able to detect the presence of and assess the relative density of bacterial patches. When foraging in environments with low-density bacterial patches, we initially observed many *searching* encounters where animals did not appear to slow down; however, over time, the proportion of *searching* encounters decreased (*Figure 2L, M*). This increased responsiveness to dilute patches over time could have resulted from small changes in bacterial growth or a modulation of patch detection sensitivity. Contrastingly, animals with reduced function of ciliated chemoreceptor and mechanoreceptor neurons due to a null mutation in the *osm-6* gene almost never *searched*. Rather, they displayed immediate *exploitation* even

while foraging on low-density patches. This suggests that *osm-6* mutants display an augmented ability to detect bacterial patches but a diminished ability to assess the bacterial density of those patches. While this may seem contradictory, it suggests that bacterial detection and density assessment are mediated by distinct sensory mechanisms. Specifically, post-ingestive feedback or other non-ciliary sensory cues may drive the initial detection of bacteria, while ciliated sensory neurons appear to be crucial for the nuanced evaluation of bacterial patch density. This interplay allows for a flexible foraging strategy where initial detection by one modality can trigger exploitation, which can then be modulated or overridden by more detailed chemosensory and mechanosensory information. This framework may explain why *osm-6* mutants, lacking the ability to finely assess bacterial density, prioritize exploitation; if all patches appear effectively equivalent, further exploration is inefficient. Collectively, these results suggest an important role for sensory modalities in the detection and assessment of bacterial patch density and highlight a need to investigate the mechanisms behind these effects in future studies.

The difficulty in assessing bacterial patch density is more than just a sensory challenge. Within-patch spatial inhomogeneity resulting from areas of active proliferation of bacteria (*Gloria-Soria and Azevedo, 2008*; *Hallatschek et al., 2007*) complicates an animal's ability to accurately assess the quantity of bacteria within a patch and, consequently, our ability to accurately compute a metric related to our assumptions of what the animal is sensing. In our study, we used the relative density of the patch edge where bacterial density is highest as a proxy for an animal's assessment of bacterial patch density (*Figure 2—figure supplement 1*). This decision was based on a previous finding that the time spent on the edge of a bacterial patch affected the dynamics of subsequent area-restricted search (*Calhoun et al., 2015*). While within-patch spatial inhomogeneity likely affects an animal's ability to assess patch density, we do not believe that this qualitatively affects the results of our study. Both the patch densities tested (*Figure 2—figure supplement 3A*) as well as our observations of time-dependent changes in *exploitation* (*Figure 2E, N, O* and *Figure 3H, I*) maintained a monotonic relationship. Therefore, alternative methods of patch density estimation should yield similar results.

According to early models in optimal foraging theory, foragers behave as if they have *complete information* about the environment (e.g., animals are assumed to know the average density of bacterial patches) (*Stephens and Krebs, 1986*). Later theoretical and empirical work expanded models of patch choice to accommodate situations where animals have *incomplete information* (e.g., environments with fluctuating resources) and, therefore, must both acquire information and forage (*Pyke et al., 1977*; *Stephens et al., 2007*; *Krebs and Inman, 1992*). This work suggests that animals may continuously sample areas in their environment to keep track of resource availability. Here, we found that animals initially choose to *sample* patches and that the duration of this period of *sampling* is correlated with the degree of mismatch between current and recently experienced and *exploited* patches (i.e., animals spend more time *sampling* as the magnitude of difference between the patch density experienced immediately before and during the experiment increases). When there is no difference between current and prior experience (i.e., animals foraging on patches of relative density 200), animals rarely *sample*. These results suggest that *C. elegans* use *sampling* as a way of re-evaluating the overall quality of patches available in the environment when a change is detected. Further, the results of our model suggest that animals likely use the information learned during *sampling* encounters to guide their decision to *explore* or *exploit* on subsequent patch encounters. We behaviorally validated this hypothesis by applying our model predictions to animals foraging in environments with multiple patch densities. Consistent with our predictions, we observed that the lower the density of recently encountered or *exploited* patches and the higher the density of the current patch, the more likely an animal would be to *exploit* (*Figure 4F*). These results suggest that *C. elegans* can learn and remember features of recent experiences and use that learned information to guide efficient decision-making. Incorporating prior experience into the decision to *explore* or *exploit* currently available food is highly beneficial when foraging in a fluctuating environment. *C. elegans* thus may have evolved this ability to maximize foraging in the wild where animals experience a boom-and-bust environment (*Frézal and Félix, 2015*). Future studies investigating the effects of varying patch density during animal development could provide additional insights into the effects of more distant experiences on the *explore–exploit* decision. However, testing this is non-trivial as maintaining stable bacterial density conditions over long timescales requires matching the rate of bacterial growth with the rate of bacterial consumption.

In addition to external sensory information, internal state signals serve a critical role in modifying decision-making in *C. elegans* (*Scheer and Bargmann, 2023*; *Matty et al., 2022*). For example, food-deprived animals are more likely to cross an aversive barrier in search of food as compared to well-fed controls (*Matty et al., 2022*; *Ezcurra et al., 2011*). Here, we find that a satiety-related signal likely drives the decision to *exploit*. Specifically, our model suggests that as animals' satiety decreases, their willingness to accept a lower quality patch increases. We confirmed this prediction by analyzing the behavior of food-deprived animals, which immediately *exploit* even when bacterial patches are very dilute (*Figure 4—figure supplement 3*). Further, we showed that this temporal signal was not likely to have been caused by transfer-induced stress (*Figure 4—figure supplement 4D*). Combined with the effect of prior experience, these results suggest that the decision to *explore* or *exploit* likely requires integration of food-related internal and external cues. Future studies should probe if additional factors related to the animal's biotic and abiotic environment (*Guisnet et al., 2021*; *Anderson et al., 2011*; *Petersen et al., 2014*; *Fraune and Bosch, 2010*; *Dirksen P et al., 2020*) as well as alternative motivations (e.g., reproduction and predatory avoidance) (*Ding et al., 2020*; *Lipton et al., 2004*; *Barrios et al., 2008*; *Matsuura et al., 2005*) further modify these foraging decisions.

## Quantitative and ethological approach to investigating decision-making

Naturalistic observations of *C. elegans* behavior are necessary to achieve a more complete cellular, molecular, and genetic understanding of how the nervous system has evolved to drive animal behavior. For example, large patches of densely seeded bacteria have primarily been used in experiments assessing *C. elegans* foraging (*White et al., 1986*; *de Bono and Maricq, 2005*; *Gray and Lissmann, 1964*; *Croll, 1975a*; *Croll, 1975b*). However, wild nematodes experience a boom-and-bust environment (*Frézal and Félix, 2015*) that may be more consistent with sparse, patchily distributed bacteria (*Iwanir et al., 2016*). Thus, just as quiescence is only observed on high-quality bacterial patches (*You et al., 2008*), additional behavioral states may be discovered when assessing foraging in dilute patchy environments. By conducting detailed analyses of ecologically inspired foraging behaviors in individual animals, we discovered that *C. elegans* make *accept–reject* decisions upon encounter with bacterial patches. By studying foraging in these environments containing small, dilute bacterial patches, we increased the frequency of opportunities for *accept–reject* decision-making (i.e., patch encounters and patch-leaving events are more frequent) and decreased the condition-specific bias toward *accept* decisions (i.e., animals are less likely to *accept* patches when moved between environments of differing bacterial density). We expect that further investigation into naturalistic behaviors could reveal additional insights into more complex behaviors and decision-making in *C. elegans* and other animals.

In this study, we developed an assay that enables observation of ecologically relevant decision-making in freely moving animals. We posit that future studies can leverage this assay to investigate the neuronal mechanisms underlying decision-making. The frequency of decision-making opportunities (i.e., numerous *accept–reject* decisions could be observed in the span of minutes) and consistency of patch choice decisions across hundreds of animals and a range of environmental conditions tested supports this claim. Further, we anticipate that the *accept–reject* decision occurs during a narrow time window – possibly at the time of encounter with the patch edge or shortly thereafter. Therefore, we expect that this assay could be adapted for experiments utilizing calcium imaging in freely moving animals (*Faumont and Lockery, 2006*; *Ji et al., 2021*; *Prevedel et al., 2014*; *Nguyen et al., 2016*; *Venkatachalam et al., 2016*; *Voleti et al., 2019*; *Hallinen et al., 2021*) to monitor neuronal activity as animals make these decisions. Additionally, the foraging task used here could be leveraged as an accumulation of evidence paradigm, which has been predominantly studied using mammalian animal models (*Davidson and El Hady, 2019*).

Finally, we used simple quantitative models to dissect and validate the components underlying decision-making. Specifically, we employed a logistic regression GLM which we adapted to accommodate our uncertainty in classification of *sensing* and *exploitation* for each encounter. While alternative models may provide more mechanistic predictions, the GLM enabled us to conduct a sequence of hypothesis tests probing the influence of food-related signals on the decision to *exploit*. Indeed, we conducted a great number of analyses involving model selection and optimization. Ultimately, we focused our study on asking whether a set of simple covariates could be used to predict foraging decisions. The GLM enabled us to identify the simplest model that could explain the observed delay in

*exploitation*. While even the null model predicted that, on average, numerous *exploratory* encounters were likely to occur before the first *exploitation*, models including satiety and prior experience significantly better predicted the extent of this delay. Further, we demonstrated that we could validate these model predictions in food-deprived animals, animals exploring multiple patch densities, and animals with sensory deficiencies. We regard our study as an initial foray into demonstrating *accept–reject* decision-making in nematodes and, while our model is a useful tool for testing the effects of past experiences, our results do not imply that *C. elegans* discretize their experiences into the parameters used in our model. The exact mechanisms and, consequently, the best model design require further investigation. For example, it is possible that the density of recently *exploited* patches is predictive of subsequent *exploitation* because post-ingestive feedback is required for the memory and/or because more distant experiences are remembered (i.e., long-term memory is involved). Future studies should seek to differentiate between these possibilities and characterize the time window of the memory. Altogether, we suggest that this quantitative and ecologically inspired approach to investigating behavior and decision-making is incredibly powerful for refining hypotheses and enables subsequent investigation into the underlying cellular, molecular, and circuit pathways in *C. elegans*, a strategy that could be adopted across species.

# Materials and methods

**Key resources table**

| Reagent type (species) or resource | Designation | Source or reference | Identifiers | Additional information |
|---|---|---|---|---|
| Strain, strain background (*Escherichia coli*) | OP50 | *Caenorhabditis* Genetics Center | WBStrain00041969 | |
| Strain, strain background (*Escherichia coli*) | OP50-GFP | *Caenorhabditis* Genetics Center | WBStrain00041972 | |
| Strain, strain background (*Caenorhabditis elegans*) | N2: wild isolate | *Caenorhabditis* Genetics Center | WBStrain00000001 | |
| Strain, strain background (*Caenorhabditis elegans*) | PR811: [osm-6(p811) V] | *Caenorhabditis* Genetics Center | WBStrain00030796 | |
| Strain, strain background (*Caenorhabditis elegans*) | TU253: [mec-4(u253) X] | *Caenorhabditis* Genetics Center | WBStrain00035037 | |
| Software, algorithm | StreamPix 8 | NorPix | RRID:SCR_015773 | |
| Software, algorithm | WormLab | MBF Bioscience | RRID:SCR_017669 | |
| Software, algorithm | MATLAB 2024a | Mathworks | RRID:SCR_001622 | |
| Software, algorithm | Photoshop 2024 | Adobe | RRID:SCR_014199 | |
| Software, algorithm | Illustrator 2024 | Adobe | RRID:SCR_010279 | |
| Software, algorithm | Code for analysis | https://github.com/shreklab/Haley-et-al–2024 | | |

## Bacterial cultures

Stock liquid cultures of the OP50 strain of *E. coli* were prepared via inoculation of a single colony in sterile lysogeny broth (LB) grown overnight at room temperature. Stock liquid cultures were subsequently stored at 4°C for up to 6 weeks.

Solutions of OP50 for each experiment were prepared via a series of dilutions in LB (*Figure 1—figure supplement 1A*). 50 ml of OP50 stock solution was centrifuged at 3000 rpm for 5 min. After removal of the supernatant, approximately 500–1000 µl of saturated liquid culture remained. The bacterial density of this saturated solution was estimated by measuring the optical density at 600 nm ($OD_{600}$) of a 1:50 dilution of the homogenized solution on a spectrophotometer (Molecular Devices SpectraMax Plus 384). The saturated solution was subsequently diluted to achieve an $OD_{600}$ of ~10 ($\mu$ = 10.22, $\sigma$ = 0.31). Measurements of the number of colony-forming units in the '10' solution estimated $13.1 \times 10^9$ cells per ml on average. Additional densities ($OD_{600}$ = {0.05, 0.1, 0.5, 1, 2, 3, 4, 5}) were prepared via dilution of the '10' solution with LB and kept on ice to prevent bacterial growth. A '0' density solution was prepared with just LB.

For all experiments, these density solutions were seeded onto cold, low-moisture nematode growth medium (NGM) plates (3% agar) to facilitate pipetting of small, circular, quick-drying patches. Unless otherwise noted, seeded plates were immediately returned to 4°C after patches dried to prevent bacterial growth. Plates were stored at 4°C for an average of 6 days before experimentation.

For experiments using an OP50 strain expressing green fluorescent protein (OP50-GFP) (*Labrousse et al., 2000*), cultures were prepared as above, but with the addition of 100 µg/ml of carbenicillin to the liquid LB. All bacterial strains used in this study are listed in the Key Resources table.

## Nematode cultures

*C. elegans* strains were maintained under standard conditions at 20°C on NGM plates (1.7% agar) seeded with stock liquid culture of OP50 (*Brenner, 1974*). All experiments were performed on young adult hermaphrodites, picked as L4 larvae the day before the experiment ($\mu = 24.6$ hr, $\sigma = 3.4$). Unless otherwise indicated, well-fed animals of the standard *C. elegans* strain N2 Bristol were used for experiments. Transgenic strains were always compared to matched controls tested in parallel on the same days. All *C. elegans* strains used in this study are listed in the Key Resources table.

## Assay preparation and recording
### Single-density, multi-patch assay

Several days before the experiment, condition and acclimation plates were prepared (*Figure 1—figure supplement 1A*). For the condition plates, a large circular arena (30 mm in diameter) made of clear transparency film (6 mil, PET) was cut using a computer-controlled cutting machine (Cricut Maker 3) and placed in the center of a 10-cm Petri dish filled with 25 ml of NGM (3% agar). These arenas function to corral animals with a low probability of escape (*Quach et al., 2024*). A pipetting template made of the same material was designed and cut with an isometric grid consisting of 19 circular holes spaced 6 mm apart from center-to-center. The pipetting template was overlaid on top of the arena and 0.5 µl of OP50 solution (OD$_{600}$ = {0, 0.05, 0.1, 0.5, 1, 2, 3, 4, 5, 10}) was pipetted into each hole. After drying, the template was immediately removed, and the Petri dish lid replaced to prevent contamination and dehydration. For two conditions ('1 (12H)' and '1 (48H)'), plates were seeded with OD$_{600}$ = 1 and then left at room temperature for 12 or 24 hr, respectively, prior to storage at 4°C. For all other conditions, plates were immediately stored at 4°C after drying. Acclimation plates were prepared by seeding NGM (3% agar) plates with one large patch of 200 µl of OD$_{600}$ = 1 grown for 24 hr at room temperature.

Approximately 24 hr before the experiment, acclimation plates and '1 (48H)' condition plates were transferred from 4°C to room temperature. After plates warmed to room temperature (~1 hr), 30–60 L4 animals were picked onto the acclimation plates. Plates were then stored at 20°C until the experiment for a combined bacterial growth time of approximately 48 hr (24 hr from freshly seeded to 4°C; 24 hr from 4°C to experiment). The combined bacterial growth time for all condition plates was approximately 1 hr ($\mu = 1.16$ hr, $\sigma = 0.23$) unless otherwise noted as '12' ($\mu = 13.63$ hr, $\sigma = 0.50$) or '48' hr ($\mu = 49.28$ hr, $\sigma = 2.94$).

On the day of the experiment, each set of condition plates was transferred from 4°C to room temperature. After 1 hr, four young adult animals were gently transferred to an empty NGM plate to limit the spread of bacteria from acclimation to condition plates. Animals were then quickly transferred to a condition plate. Animals were transferred using the flat surface of cylindrical plugs excised from clean 3% agar (*Quach and Chalasani, 2022*; *Quach et al., 2024*). Importantly, the use of agar as a medium to transfer animals provides minimal disruption to their environment as all physical properties (e.g., temperature, humidity, and surface tension) are maintained. Qualitatively, we observe no marked change in behavior from before to after transfer with the agar plug method, especially as compared to the often drastic changes observed when using a metal or eyelash pick.

Condition plates were subsequently placed face down on a piece of glass suspended above an edge-lit backlight (Advanced Illumination). Recordings were acquired using PixeLink cameras (PL-B741F) combined with Navitar lenses (1-60135 and 1-6044) and Streampix 8 software. Behavior was recorded for 1 hr at 1024 × 1024 pixels and 3 frames per second (fps) with spatial resolution of ~33 pixels per mm ($\mu = 32.68$ pixels/mm, $\sigma = 1.70$).

The assay was repeated over numerous days with every condition assayed on each day when possible. The order of seeding assay plates and recording behavior was randomized for each day to

compensate for the potential effects of time on animal age and bacterial density. Further, the average temperature ($\mu = 21.99°C$, $\sigma = 0.95$) and humidity ($\mu = 51.39\%$, $\sigma = 11.67$) during each experiment were recorded. As a result of randomization, animal age, bacterial growth time, temperature, and humidity did not significantly vary between density conditions. Replication of the assay was constrained by the setup (~4 hr per day), recording (1–2 hr per animal), and analysis (0.5–1 hr per animal) time required. A maximum of 24 animals (2 cameras run for 12 hr) could be assayed per day. Given these resource constraints, we collected replicates until saturation of qualitative observations (i.e., new animals merely replicated earlier observations without adding new information) (*Morse, 1995*) resulting in sample sizes comparable to similar studies (*Iwanir et al., 2016*; *Pradhan et al., 2019*).

### Large, single-patch assay

Plates were prepared as described for the single-density, multi-patch assay with the exception that only one large patch was created by pipetting 20 µl of OP50 solution ($OD_{600} = \{0, 0.05, 0.1, 0.5, 1, 2, 3, 4, 5, 10\}$) into the center of the 30-mm diameter arena.

### Small, single-patch assay

Plates were prepared as described for the single-density, multi-patch assay with the exception that only one small patch was created by pipetting 0.5 µl of OP50 solution ($OD_{600} = \{0, 0.5, 1, 5, 10\}$) into the center of a 9-mm diameter arena. Given the smaller arena size, only one young adult animal was transferred into the arena. Behavior was recorded for 1 hr at higher spatial ($\mu = 104.76$ pixels/mm, $\sigma = 17.6$) and temporal resolution (8 fps). The acquisition setup was otherwise unaltered except for the removal of a ×0.25 Navitar lens (1-6044) from the light path.

### Multi-density, multi-patch assay

Plates were prepared as described for the single-density, multi-patch assay with the exception that varying combinations of $OD_{600} = \{1, 5, 10\}$ (i.e., $OD_{600} = \{1, 5, 10, 1 + 5, 1 + 10, 1 + 5 + 10\}$) were pipetted onto each assay plate in a patterned isometric grid of 18 0.5 µl droplets. Behavior was recorded for 2 hr. To compensate for the longer duration recordings, NGM agar plates were poured without Bacto peptone (BD 211677) to prevent bacterial growth. The result of this change was that the relative density of bacterial patches was ~30% lower as compared to comparable 0.5 µl patches in the single-density, multi-patch assay and the small, single-patch assay (*Figure 2—figure supplements 2 and 3*). Further, to prevent censorship of our data at the beginning of the recording, we formed a small droplet of S-Complete solution (*Sulston and Brenner, 1974*) in the middle of the arena where no bacterial patch was placed. Animals were transferred to this droplet using an eyelash pick immediately prior to the assay plate being placed on the imaging setup. The recording was started once the droplet evaporated and animals were free to crawl about the arena. This process ensured that recordings began prior to an animal's first patch encounter.

## Bacterial patch location detection

Most bacterial densities tested in these experiments were too dilute to be visible under normal imaging conditions. Therefore, several strategies were employed to accurately detect the location of bacterial patches. First, a small reference dot was cut into the arena and pipetting templates enabling consistent and traceable patch placement. Further, prior to each behavior recording, a 'contrast' video was acquired wherein a piece of dark cardstock was passed between the light source and the condition plate (*Figure 1—figure supplement 1B–D*, *Video 2*). This produced an effect where diffraction of light through the patches provided enough contrast to view the patches. Binary masks of the circular arena and patches were extracted from these videos in MATLAB using the Image Processing Toolbox and further refined manually in Photoshop (Adobe, 2024). Arena masks were used to calculate the scale (pixels/mm) of each image. In early experiments, the 'contrast' video was acquired prior to worms being added to the condition plate, which resulted in displacement of the arena within the camera's field of view. Image registration using MATLAB's Computer Vision Toolbox was performed to accurately map the patches detected in the 'contrast' video onto the behavioral recording.

## Bacterial patch density estimation

As described above (see Assay preparation and recording), we varied the relative density of bacterial solutions by diluting OP50 in LB and growing patches for different lengths of time. As a result of these procedures, the relative bacterial density throughout the assay was not known and needed to be estimated. To determine the relative density of these bacterial patches, we seeded plates with fluorescently labeled OP50-GFP under identical conditions as in our assays (*Figure 2—figure supplement 1A*). We imaged these bacterial patches for several hours using a Zeiss Axio Zoom.V16. Images were analyzed using the Image Processing Toolbox in MATLAB. After correcting for inconsistent illumination across the field of view and normalizing images using matched controls (*Figure 2—figure supplement 1B, C*), we detected the location of each bacterial patch and extracted a fluorescence intensity profile along each patch's radius (*Figure 2—figure supplement 1D*). Consistent with previous studies, we found that even in extremely dilute conditions, bacterial density was greater at the patch edge where actively proliferating bacteria are concentrated (*Gloria-Soria and Azevedo, 2008*; *Hallatschek et al., 2007*). Therefore, we detected the patch edge and computed the peak amplitude of each bacterial patch at all time points assayed (*Figure 2—figure supplement 1E*). We then performed linear regression on the values for each condition to create models of peak amplitude as a function of time. For low-density conditions (i.e., $OD_{600} = \{0.05, 0.1\}$) of small patches, fluorescent signals could not be detected. We therefore estimated these functions from a multinomial regression of values taken from $OD_{600} = \{0.5, 1, 2\}$. Through this process, we used the total time that bacteria were allowed to grow at room temperature for each condition plate to estimate the peak amplitude at each time point (*Figure 2—figure supplement 3*). Finally, to minimize confusion about density conditions between experiments, we computed values for relative density by linearly scaling estimated peak amplitude values so that 0.5 µl patches seeded with $OD_{600} = 10$ and grown for 1 hr at room temperature would have relative density equal to 10. We labeled each condition using the relative density rounded to one significant digit.

## Behavioral tracking

WormLab (MBF Bioscience) was used for tracking animal behavior (*Figure 1—figure supplement 1E–G*, *Video 3*). For each video, we adjusted thresholds for pixel intensity and worm dimensions to enable automatic tracking. WormLab then fit the worm's body at every frame and stitched together a track of each worm across frames. Subsequent manual corrections were required for instances where the software created new worm tracks. This occurred most frequently when animals: (1) encountered the arena border, (2) collided with each other, or (3) overlapped with bubbles or dust particles embedded in the agar. To correct for these discontinuities, we manually 'stitched' worm tracks together enabling us to keep track of each animal's location throughout the duration of the experiment. We exported the location of each animal's midpoint for further analysis in MATLAB (Mathworks, 2024a). For small, single-patch assays where higher spatial resolution enabled reliable distinction between the head and tail of the animal, we manually confirmed the head–tail and exported 25 points along the animal's midline (*Figure 1—figure supplement 2A–C*).

Animals were excluded from analyses when (1) the animal's location could only be tracked for less than 75% of the video due to the animal escaping the arena and when (2) the location of bacterial patches could not be reliably assessed due to missing or low-quality 'contrast' videos (see Bacterial patch location detection).

## Patch encounter detection

As the head position of animals in our experiments was difficult to track due to low spatial resolution and limited automated detection of head position using WormLab, we instead tracked the midpoint position of the worm's body and estimated patch encounters using a set of measured criteria. To do this, we tracked the behavior of worms in higher resolution (~105 pixels/mm compared to ~33 pixels/mm and 8 fps compared to 3 fps), single patch assays (*Figure 1—figure supplement 2A–C*). We found that when the head of the animal was in contact (i.e., within 1 pixel) with the patch edge (see Bacterial patch location detection), the animal's midpoint was on average 0.46024 mm away (*Figure 1—figure supplement 2D*). We therefore defined a patch encounter as the time when an animal's midpoint came within 0.46024 mm of the patch edge. However, this definition led to two

types of errors: (1) putative entry events where the animal merely passed by the patch and (2) putative leaving events where the animal did not fully leave the patch.

To remove putative entry events where the animal nearly passed by the patch, we defined an additional distance threshold based on the distance between the animal's midpoint and the patch edge for all time points when the head was within the patch (*Figure 1—figure supplement 2E*). The midpoint was almost always (99% of the time) within 0.28758 mm of the patch edge. Therefore, we removed patch encounters where the midpoint never got closer than 0.28758 mm from the patch edge. Although this criterion removes most 'near-miss' events, some events remain where the animal approached the patch, but its head never entered. Further decreasing this threshold (e.g., midpoint must be on patch) would result in real events being excluded due to the animal taking on specific postures where the midpoint remains outside the patch while feeding within the patch. Therefore, we chose to be more liberal (i.e., more false positive patch encounters than false negatives) in our detection of patch encounters. Subsequent analysis of these putative patch encounters removed the majority of remaining false positives (see Patch encounter classification as sensing or non-responding).

To address putative leaving events and avoid splitting a single patch encounter into multiple encounters, we considered that a lawn leaving event is most frequently defined as an event where all body parts have left the patch (*Scheer and Bargmann, 2023*). When tracking only the midpoint, it is not possible to ensure that the worm's entire body has exited the patch. Using the criterion that an animal exits a patch when its midpoint exceeds 0.46024 mm from the patch edge accurately predicts exit events most of the time, but under certain scenarios (e.g., animals maintain an outstretched feeding posture *Quach and Chalasani, 2022*) false leaving events were detected. To exclude these leaving events, we analyzed the variability in the distance from the midpoint to the patch edge during on- and off-patch events (*Figure 1—figure supplement 2F*). We found that on-patch events had low distance variability while off-patch events displayed a bimodal distribution, with some events having high variability as expected, and some events having low variability. We fit a two-component GMM to the off-patch distance variability and found that low variability leaving events could be reliably excluded when the standard deviation of off-patch distances was less than 0.22221. We subsequently combined putative patch encounters where variability was below this threshold, resulting in fewer overall encounters.

## Patch encounter classification as exploration or exploitation

To classify patch encounters as *exploration* or *exploitation* events, patch encounters were detected (see Patch encounter detection) and two features were computed: (1) duration of patch encounter and (2) average on-patch velocity. Both the duration and velocity features displayed a bimodal distribution with two visible clusters: one with short-duration encounters and high on-patch velocity and one with long-duration encounters and low on-patch velocity (*Figure 2—figure supplement 6A*). To confirm the existence of two clusters, we conducted a modified version of Silverman's test for bimodality (*Ahmed and Walther, 2012*; *Silverman, 1981*). Broadly, Silverman's test uses kernel density estimation (KDE) to investigate the number of modes in a distribution. By solving for a critical bandwidth at which the KDE switches modality from $j$ to $j + 1$ modes, one is able to test the null hypothesis that a distribution has $j$ modes, versus the alternative that it has more than $j$ modes. Following a previously described protocol (*Ahmed and Walther, 2012*; *Silverman, 1981*), we tested the null hypothesis that our data set was unimodal (i.e., $j = 1$). We first mean-centered the matrix $z_k$ containing the log transforms of the duration and average velocity of each patch encounter $k$. We then calculated the projection of each data point onto the first principal component (PC1) of the data to get a one-dimensional distribution representing the data in $z_k$ (*Figure 2—figure supplement 6B*). The use of principal *components* instead of principle *curves* as described previously was justified as both methods are indistinguishable when $j = 1$ (*Ahmed and Walther, 2012*). Two modes are visible in both the histogram and KDE of the distribution of PC1 projected data when the KDE is computed using a Gaussian kernel and Silverman's Rule of Thumb for bandwidth (*Silverman, 1986*) defined as $\hat{h} = \sigma \left( \frac{4}{(d+2)\, n} \right)^{\frac{1}{d+4}}$ where, for this data set, $\sigma$ is the standard deviation, $d = 1$ is the dimensionality, and $n = 6560$ is the number of observations. To test the significance of this distribution being bimodal rather than unimodal, we computed the critical bandwidth $h^*$ defined as the minimum bandwidth for which the KDE has only one mode. Following Silverman's test procedure, we computed a smoothed bootstrap across 2000 replicates by sampling the data with replacement and adding noise such that

for any data point $z_i$ and bootstrap resample $b$, we make noisy $\widetilde{z}_{i,b} = \frac{z_i + h^* \epsilon_{i,b}}{\sqrt{1 + \left(\frac{h^*}{\sigma}\right)^2}}$ where $\epsilon_{i,b}$ is a standard normal random variable independently drawn for each $i$ and $b$. We found that the critical bandwidth $h^*$ of our data set was significantly (p < 0.001) greater than that of the bootstrapped data sets, which allows us to reject the null hypothesis that the distribution of $z_k$ is unimodal and justifies the use of a two-cluster model in classifying these data (*Figure 2—figure supplement 6C*).

After confirming the bimodality of the data set, we classified patch encounters using a two-component GMM to estimate $p\left(y_k = 1|z_k\right)$ where for each encounter $k$, $z_k$ are the log transforms of the duration and average velocity of the patch encounter, $y_k = 1$ indicates an *exploitation*, and $y_k = 0$ indicates an *exploration* (*Figure 2—figure supplement 6D*). We optimized the GMM across cross-validated replicates with respect to the regularization value $\alpha$ to minimize the posterior variance given by

$$\sum_k p\left(y_k = 1|z_k\right) p\left(y_k = 0|z_k\right).$$

We found that $\alpha = 0.025$ best separated the *exploration* and *exploitation* clusters (*Figure 2—figure supplement 6E*). The posterior probability of *exploitation* $p\left(y_k = 1|z_k\right)$ was subsequently estimated for all encounters (*Figure 2—figure supplement 6F, G*).

## Patch encounter classification as sensing or non-responding

To classify patch encounters as *sensing* or *non-responding* events, we computed animals' (1) deceleration upon encounter with the patch edge, (2) minimum velocity during the patch encounter, and (3) maximum change in velocity between the peak velocity immediately prior to the start of the patch encounter and the minimum velocity during the encounter (*Figure 2—figure supplement 7A, B*). Deceleration was defined as the slope of the line fit on an animal's velocity between –1.5 s before and 6.5 s after the start of an encounter. Minimum velocity was defined as the absolute minimum velocity observed during the duration of the patch encounter. The maximum change in velocity was computed by subtracting the minimum velocity from the peak velocity within 10 s of the patch encounter. The combination of these three metrics reveals two non-Gaussian clusters confirmed by Silverman's test (p = 0.003) as described in the previous section (*Figure 2—figure supplement 7C–F*). These clusters represent encounters where animals sensed the patch (i.e., slow minimum velocity, large changes in velocity) or did not respond to it (i.e., maintained fast minimum velocity, with little to no change).

We estimated the probability of *sensing* $p\left(v_k = 1|w_k\right)$ where for each encounter $k$, $w_k = (s_k, t_k, u_k)$ are the three velocity-related features, $v_k = 1$ indicates an encounter that was *sensed*, and $v_k = 0$ indicates an encounter where *no response* was detected. Using a semi-supervised approach to QDA, we labeled a subset of all encounters and iteratively estimated labels on the remaining unlabeled data (*Figure 2—figure supplement 8A–C*, *Video 5*). To label the data, we made two simple assumptions: (1) animals must have sensed the patch if they exploited it and (2) animals must not have sensed the patch if there was no bacteria to sense. Therefore, all encounters with bacteria-free patches (i.e., LB only, relative density 0) were labeled as true negatives $v_k = 0$ and, across 1000 replicates, we probabilistically included *exploitation* encounters $y_k = 1$ estimated from the distribution $y_k|z_k \sim \text{Bern}\left(p\left(y_k = 1|z_k\right)\right)$ as true positives $v_k = 1$. The semi-supervised QDA method then used these initial labels to iteratively fit a paraboloid that best separated the labeled data, by minimizing the posterior variance of classification. Using this approach, the conditional probability of *sensing* $p\left(v_k = 1|w_k\right)$ was estimated for all encounters using QDA and averaged across replicates (*Figure 2—figure supplement 8D–E*).

Although we found that the two clusters were successfully discriminated by this semi-supervised QDA approach, a small subset of encounters (252 of 20109 encounters) could not be reliably classified in this manner as the onset of the patch encounter was not observed. This type of data censoring occurred when the behavioral recording was started after the animal had already entered the patch. Although we could compute deceleration-related metrics for these encounters with the assumption that the animal entered the patch at the first frame, these measurements are censored with a bias toward lower magnitudes of deceleration. Therefore, to better predict the conditional probabilities of *sensing* for these 252 encounters, we assumed that the minimum velocity on-patch $s_k$ was a reliable

metric and marginalized the conditional probabilities over the other two metrics ($t_k$ and $u_k$) as defined by

$$p\left(v_k = 1|s_k\right) = \int_{-\infty}^{\infty} \int_{-\infty}^{\infty} p\left(v_k = 1|s_k, t_k, u_k\right) \, p\left(t_k, u_k \,|s_k\right) dt \, du.$$

We numerically integrated using an adaptive quadrature method over the product of the QDA estimated conditional probability distribution and the KDE of the joint probability distribution. This procedure resulted in a slight increase ($\mu = +0.0615$, $\sigma = 0.1832$) in our estimation of the probability of *sensing* (*Figure 2—figure supplement 8F*).

Altogether, this classification approach resulted in low false positive and false negative rates. Specifically, 3.35% of the time encounters with bacteria-free patches were incorrectly identified as *sensing*, while 2.92% of the time *exploitatory* encounters were incorrectly identified as *non-responding*. Further, this classification removed many of the remaining near-miss patch encounters in which animals came close to but did not truly enter the patch (see Patch encounter detection). For all analyses, we excluded near-miss encounters where the animal's midpoint never entered the patch and the probability of *sensing* was less than 5% (i.e., $p\left(v_k = 1|w_k\right) < 0.05$).

## Models of exploitation probability

We consider that the probability of *exploiting* a patch upon any given encounter can be described by a logistic function:

$$p\left(y_k = 1|\boldsymbol{\beta} \cdot \boldsymbol{x}_k\right) = \frac{1}{1 + e^{-\boldsymbol{\beta} \cdot \boldsymbol{x}_k}},$$

where $p\left(y_k = 1|\boldsymbol{\beta} \cdot \boldsymbol{x}_k\right)$ represents the conditional probability that an animal exploited during patch encounter $k$; $\boldsymbol{x}_k \in R^n$ is a vector of covariates; and $\boldsymbol{\beta} \in R^n$ is a vector of weights that describes how much each covariate influences the animal's choice. In models compared here, $\boldsymbol{x}_k$ includes a combination of a constant element as well as covariates that may empirically influence the decision to exploit: (1) the log-transformed relative density of the encountered patch $k$ ($\rho_k$), (2) the duration of time spent off food since departing the last *exploited* patch ($\tau_s$), (3) the log-transformed relative density of the patch encountered immediately before encounter $k$ ($\rho_h$), and (4) the log-transformed relative density of the last *exploited* patch ($\rho_e$).

The observations of *exploitation* and *sensation* for our data are derived predictions from classification models rather than direct measurements (see Patch encounter classification as exploration or exploitation and Patch encounter classification as sensing or non-responding). As such, the labels for both whether a patch encounter was *sensed* by the animal $p\left(v_k = 1|w_k\right)$ and whether the animal *exploited* the patch $p\left(y_k = 1|z_k\right)$, are both reported as probabilities rather than binary, deterministic variables.

To account for uncertainty in *sensation*, we produced 100 sets of 'encounter sampled' observations wherein we probabilistically included *sensed* encounters $v_k = 1$ estimated from the distribution $v_k|w_k \sim \text{Bern}\left(p\left(v_k = 1|w_k\right)\right)$. In doing so, we assume that only patch encounters recognized by the animal guide the decision to *exploit* (*Figure 4B*). After removing *non-responding* encounters where $v_k = 0$, we defined observations of all covariates (i.e., $\rho_k$, $\tau_s$, $\rho_h$, and $\rho_e$) using only *sensed* encounters.

To account for uncertainty in *exploitation*, we modified the standard maximum likelihood. To understand our approach, consider that standard logistic regression models binary observations as realizations of the Bernoulli distribution $y_k|x_k \sim \text{Bern}\left(p\left(y_k = 1|\boldsymbol{\beta} \cdot \boldsymbol{x}_k\right)\right)$ and that the likelihood for this model is given by

$$\prod_k p\left(y_k = 1|\boldsymbol{\beta} \cdot \boldsymbol{x}_k\right)^{y_k} \left(1 - p\left(y_k = 1|\boldsymbol{\beta} \cdot \boldsymbol{x}_k\right)\right)^{1 - y_k}.$$

However, we do not have direct observations of $y_k$ and cannot evaluate the logistic likelihood directly. Rather, we have posterior probabilities of *exploitation* determined by a classifier, denoted by $p\left(y_k = 1|z_k\right)$, where $z_k$ are a set of velocity-related features that are distinct from $\boldsymbol{x}_k$ (see Patch encounter classification as exploration or exploitation). Thus, rather than maximizing the Bernoulli likelihood, we accommodate our uncertainty about $y_k$ by learning the regression parameters that minimize the Kullback–Leibler (KL) divergence between $p\left(y_k|\boldsymbol{\beta} \cdot \boldsymbol{x}_k\right)$ and the reference probability distribution $p\left(y_k|z_k\right)$ as given by

$$D_{KL} \left[ p\left(y_k|z_k\right) \| p\left(y_k|\boldsymbol{\beta} \cdot \boldsymbol{x}_k\right) \right] \equiv \sum_{y_k} p\left(y_k|z_k\right) \log \left[ \frac{p\left(y_k|z_k\right)}{p\left(y_k|\boldsymbol{\beta} \cdot \boldsymbol{x}_k\right)} \right]$$

$$= C_{\boldsymbol{\beta}} - \sum_{y_k} p\left(y_k|z_k\right) \log \left[ p\left(y_k|\boldsymbol{\beta} \cdot \boldsymbol{x}_k\right) \right]$$

$$= C_{\boldsymbol{\beta}} - \sum_{y_k} p\left(y_k|z_k\right) \log \left[ p\left(y_k|\boldsymbol{\beta} \cdot \boldsymbol{x}_k\right)^{y_k} \left(1 - p\left(y_k|\boldsymbol{\beta} \cdot \boldsymbol{x}_k\right)\right)^{1-y_k} \right]$$

$$= C_{\boldsymbol{\beta}} - p\left(y_k = 1|z_k\right) \log \left[ p\left(y_k|\boldsymbol{\beta} \cdot \boldsymbol{x}_k\right) \right]$$
$$\quad - \left(1 - p\left(y_k = 1|z_k\right)\right) \log \left[ 1 - p\left(y_k|\boldsymbol{\beta} \cdot \boldsymbol{x}_k\right) \right]$$

$$= C_{\boldsymbol{\beta}}$$
$$\quad - \log \left[ p\left(y_k|\boldsymbol{\beta} \cdot \boldsymbol{x}_k\right)^{p(y_k=1|z_k)} \left(1 - p\left(y_k|\boldsymbol{\beta} \cdot \boldsymbol{x}_k\right)\right)^{1-p(y_k=1|z_k)} \right],$$

where $C_{\beta}$ is a term that is constant with respect to $\beta$, and the second term is equivalent to the logarithm of the logistic likelihood with $p\left(y_k = 1|z_k\right)$ replacing $y_k$. Put plainly, minimizing the sum of the KL divergence between our classifier probabilities and logistic regression probabilities over all encounters $k$ is equivalent to maximizing the logistic likelihood with our classifier-based *exploitation* probabilities $p\left(y_k = 1|z_k\right)$ as observations instead of direct measurements of *exploitation* $y_k$ as given by

$$\sum_k \log \left[ p\left(y_k|\boldsymbol{\beta} \cdot \boldsymbol{x}_k\right)^{p(y_k=1|z_k)} \left(1 - p\left(y_k|\boldsymbol{\beta} \cdot \boldsymbol{x}_k\right)\right)^{1-p(y_k=1|z_k)} \right].$$

Notably, when there is no observation uncertainty, $p\left(y_k = 1|z_k\right)$ collapses to a delta function on $y_k = 1$ and we recover the standard likelihood function. Thus, our method is a generalization of maximum likelihood. To train our model, we therefore fit a standard logistic regression GLM where we provided sets of observations of covariates $\boldsymbol{x}_k$ and response variables $p\left(y_k = 1|z_k\right)$ for *sensed* encounters.

The covariates $\boldsymbol{x}_k$ included the relative density of the encountered patch $k$ ($\rho_k$), the duration of time spent off food since departing the last *exploited* patch ($\tau_s$), the relative density of the patch encountered immediately before encounter $k$ ($\rho_h$), and the relative density of the last *exploited* patch ($\rho_e$). For every encounter $k$, $\rho_k$ is the log-transformed relative density of the encounter patch as estimated from experiments using a fluorescent bacterial strain, OP50-GFP (as described in Bacterial patch density estimation). $\tau_s$ is the duration of time spent off food since the beginning of the recorded experiment (i.e., total time elapsed minus duration of time on patch). For the first patch encounter (i.e., $k = 1$), this is equivalent to the total time elapsed. $\rho_h$ is the log-transformed relative density of the patch encountered immediately before encounter $k$. For the first patch encounter, we initialized this parameter using the approximated relative density of the bacterial patch on the acclimation plates (see Assay preparation and recording in Methods). Acclimation plates contained one large 200 µl patch seeded with $OD_{600} = 1$ and grown for a total of ~48 hr. The relative density of these acclimation plates was estimated using the same procedure used to estimate values of $\rho_k$ (see Bacterial patch density estimation). $\rho_e$ is the log-transformed relative density of the most recently *exploited* patch. For the first patch encounter as well as every encounter prior to the first observed *exploitation*, we assume that animals must have *exploited* within the last 24 hr while on the acclimation plates and thus initialized $\rho_e$ using the approximated relative density of the bacterial patch on the acclimation plates.

## Exploitation of single-density, multi-patch environments

As described above, we used an 'encounter sampling' protocol to generate 100 replicate observations of only *sensed* encounters by removing all encounters where *sensing* could not be determined (i.e., $v_k = 0$) as randomly estimated from the distribution $v_k|\boldsymbol{w}_k \sim \text{Bern}\left(p\left(v_k = 1|\boldsymbol{w}_k\right)\right)$. For example, we can consider an animal foraging in the single-density, multi-patch environment (**Figure 4B**). Although we detected 10 total encounters $K = k_1, k_2, ..., k_{10}$, several of these encounters had low probabilities of *sensing* $p\left(v_k = 1|\boldsymbol{w}_k\right)$ (**Figure 4B**). We use our 'encounter sampling' protocol to simulate the *sensation* of the patch during each of these 10 encounters and find that $v_k = 0$ for $k_1, k_3, k_6$ and $k_8$. By removing these four no response encounters, we generate a new observation comprised of the six *sensed* encounters $K_1^* = k_2, k_4, k_5, k_7, k_9, k_{10} = k_{1*}, k_{2*}, ..., k_{6*}$. Using only the data related to this new sequence of

encounters $K_1^*$, we generated a vector of covariates (i.e., $\rho_k$, $\tau_s$, $\rho_h$, and $\rho_e$). Importantly, this procedure considers time spent during the *no response* encounters $k_1, k_3, k_6$, and $k_7$ the same as time spent searching off patch. We repeated this process 100 times to generate new observations of encounter sequences (i.e., $K_1^*, K_2^*, ..., K_{100}^*$). In doing so, we reduced the number of encounters from 6560 total encounters (as defined in Patch encounter detection) to 2604–2698 ($\mu = 2659.8$, $\sigma = 15.9$) *sensed* encounters for each of the 100 replicates.

Additionally, we used a 'worm sampling' protocol to generate 500 hierarchically bootstrapped replicates (*Saravanan et al., 2020*). Specifically, we took our vector of 443 animals $A = a_1, a_2, ..., a_{443}$ and resampled with replacement (e.g., $A_1^* = a_{317}, a_{88}, ..., a_{130} = a_{1*}, a_{2*}, ..., a_{443*}$) to generate 500 new samples $A_1^*, A_2^*, ..., A_{500}^*$. When combined, the 'encounter sampling' and 'worm sampling' protocols created 50,000 unique replicates of observations of the covariates $\boldsymbol{x}_k$ and response variables $p\left(y_k = 1|z_k\right)$ which were used to estimate the coefficients $\beta$ (*Figure 4C*). A null distribution of coefficients $\beta^*$ was estimated by shuffling the response variable vector (*Figure 4—figure supplement 1C*). A two-tailed, one-sample bootstrap hypothesis test was used to assess whether our covariate estimates were significantly greater than or less than 0 (i.e., $\boldsymbol{p} = 2 \times \min\left[P\left(\boldsymbol{\beta} \leq 0\right), P\left(\boldsymbol{\beta} \geq 0\right)\right]$).

## Exploitation of food-deprived animals in single-density, multi-patch environments

To validate model predictions related to the satiety signal covariate $\tau_s$, a novel data set of well-fed and 3-hr food-deprived animals foraging in single-density, multi-patch environments of relative density 5 was generated (*Figure 4—figure supplement 3A, B*). The 'encounter sampling' protocol described above was used to generate 100 replicates of observations of the covariates $\boldsymbol{x}_k$ and response variables $p\left(y_k = 1|z_k\right)$ for this data set. As a result of 'encounter sampling', we reduced the number of encounters from 42 total encounters to 30–37 ($\mu = 34.6$, $\sigma = 1.3$) *sensed* encounters for food-deprived animals and 235 total encounters to 94–107 ($\mu = 99.8$, $\sigma = 2.1$) *sensed* encounters for well-fed animals. Coefficient values $\beta$ were re-estimated in the absence of the satiety signal $\tau_s$ (*Figure 4—figure supplement 3C*) using the original single-density, multi-patch data set. Using the mean coefficient values $\beta$ for both models (with and without satiety), we predicted the conditional probabilities of exploiting $p\left(y_k = 1|\boldsymbol{\beta} \cdot \boldsymbol{x}_k\right)$ for the new data set for well-fed and food-deprived animals. We subsequently simulated a series of *exploitation* events from a Bernoulli distribution using the estimated (i.e., $y_k|\boldsymbol{x}_k \sim \text{Bern}\left(p\left(y_k = 1|\boldsymbol{\beta} \cdot \boldsymbol{x}_k\right)\right)$) and observed (i.e., $y_k|z_k \sim \text{Bern}\left(p\left(y_k = 1|z_k\right)\right)$) *exploitation* probabilities. These simulated *exploitations* were used to generate distributions of the probability of an *exploitation* occurring for the first time as a function of the number of encounters (*Figure 4—figure supplement 3D*).

## Exploitation of multi-density, multi-patch environments

To investigate the individual contributions of a satiety-related signal and transfer-induced stress, a novel data set of animals foraging on multi-density, multi-patch environments was generated (*Figure 4E*, *Figure 4—figure supplement 4A–C*). The 'encounter sampling' protocol described above was used to generate 100 replicates of observations of the covariates $\boldsymbol{x}_k$ and response variables $p\left(y_k = 1|z_k\right)$ for this data set. As a result of 'encounter sampling', we reduced the number of encounters from 4296 total encounters (350–984 total encounters per condition) to 1702–1745 ($\mu = 1724.6$, $\sigma = 9.9$) *sensed* encounters. Coefficient values $\beta$ were estimated for models including combinations of $\tau_s$, the duration of time an animal spent searching off-food since the last *exploitation* event, and $\tau_t$, the duration of time since transfer (i.e., the time elapsed in the experiment) (*Figure 4—figure supplement 4D*) using the new multi-density, multi-patch data set. While all coefficients were significant in models containing only $\tau_s$ or $\tau_t$, only $\tau_s$ was significant in a model containing both terms (i.e., $p\left(y_k = 1|\beta_0 + \beta_k\rho_k + \beta_s\tau_s + \beta_t\tau_t + \beta_h\rho_h + \beta_e\rho_e\right)$).

Further, to validate model predictions related to the prior experience covariates $\rho_h$ and $\rho_e$, coefficient values $\beta$ were re-estimated in the absence of the recently encountered and *exploited* patch density terms $\rho_h$ and $\rho_e$ (*Figure 4—figure supplement 4E*) using the original single-density, multi-patch data set. Using the mean coefficient values $\beta$ for both models (with and without prior experience), we predicted the probabilities of *exploiting* $p\left(y_k = 1|\boldsymbol{\beta} \cdot \boldsymbol{x}_k\right)$ for the new data set. To generate the heat maps of predicted behavior (*Figure 4F*), we varied $\rho_k$ and $\rho_h$ as well as $\rho_k$ and $\rho_e$ across a range of values and set the remaining covariate terms to the average values observed in the data set. To generate the heat maps of observed behavior (*Figure 4F*), we averaged the observed probabilities

of *exploiting* $p\left(y_k = 1|z_k\right)$ for each pairing of $\rho_k$ and $\rho_h$ as well as $\rho_k$ and $\rho_e$ and linearly interpolated values between these nine points.

## Exploitation of sensory mutants in single-density, multi-patch environments

To test the utility of our model in identifying covariate-specific phenotypes in animals with varied genotypes, a novel data set of wild-type animals and sensory mutants foraging on single-density, multi-patch environments was generated (*Figure 4—figure supplement 5A, B*). The 'encounter sampling' and 'worm sampling' protocols described above were used to generate 50,000 replicates of observations of the covariates $x_k$ and response variables $p\left(y_k = 1|z_k\right)$ for this data set. As a result of 'encounter sampling', we reduced the number of encounters from 1352 total encounters (27–193 total encounters per condition) to 862–894 ($\mu = 879.4$, $\sigma = 7.4$) *sensed* encounters. As the number of encounters was significantly less in these data sets as compared to the original single-density, multi-patch data set, we used ridge regression to regularize the magnitude of the covariates. We optimized each model across cross-validated replicates by varying the regularization value $\lambda$. We found that $\lambda_{\mathrm{N2}} = 0.0175$, $\lambda_{mec-4} = 0.0223$, and $\lambda_{osm-6} = 0.0614$ maximized the mean log-likelihood of each model (*Figure 4—figure supplement 5E*). Coefficient values $\beta$ were estimated for each of the three strains tested using these values (*Figure 4—figure supplement 5D*). To assess whether the magnitude of coefficients significantly differed between wild-type and mutant models, we conducted a two-sample test of the mean of differences as given by

$$Z = \frac{\mu_{\mathrm{mutant}} - \mu_{\mathrm{N2}}}{\sqrt{\sigma_{\mathrm{mutant}}^2 + \sigma_{\mathrm{N2}}^2}},$$

where $\mu$ and $\sigma$ are the mean and standard deviation, respectively, of the distribution of a covariate across replicates. Statistical significance was calculated as a left-tailed test $\phi\left(Z\right)$ for the density coefficients ($\beta_k$, $\beta_h$, and $\beta_e$) and a two-tailed test $\phi\left(-\left|Z\right|\right)$ for the other coefficients ($\beta_0$ and $\beta_s$) where $\phi$ is the standard normal cumulative distribution function.

## Acknowledgements

We thank members of the Chalasani lab for comments on the manuscript and James Fitzgerald for advice on modeling. This research was funded by grants from the National Science Foundation (J.A.H.), UCSD Undergraduate Summer Research Award (T.C.), NIH RO1 MH096881, Dorsett Brown Foundation, and Salk Innovation Grant (S.H.C.).

## Additional information

### Funding

| Funder | Grant reference number | Author |
| --- | --- | --- |
| National Institutes of Health | RO1 MH096881 | Sreekanth H Chalasani |
| Dorsett Brown Foundation | | Jessica A Haley<br>Sreekanth H Chalasani |
| Salk Institute for Biological Studies | Innovation Grant | Jessica A Haley<br>Sreekanth H Chalasani |
| National Science Foundation | Graduate Research Fellowship Program DGE-1650112 | Jessica A Haley |
| University of California, San Diego | Undergraduate Summer Research Award | Tianyi Chen |

The funders had no role in study design, data collection, and interpretation, or the decision to submit the work for publication.

## Author contributions
Jessica A Haley, Conceptualization, Resources, Data curation, Software, Formal analysis, Supervision, Funding acquisition, Validation, Investigation, Visualization, Methodology, Writing – original draft, Project administration, Writing – review and editing; Tianyi Chen, Data curation, Investigation, Writing – review and editing; Mikio Aoi, Formal analysis, Supervision, Visualization, Methodology, Writing – review and editing; Sreekanth H Chalasani, Conceptualization, Resources, Supervision, Funding acquisition, Writing – review and editing

## Author ORCIDs
Jessica A Haley ⬥ https://orcid.org/0000-0001-6282-7124
Tianyi Chen ⬥ https://orcid.org/0009-0003-7572-2155
Mikio Aoi ⬥ https://orcid.org/0000-0002-7052-880X
Sreekanth H Chalasani ⬥ https://orcid.org/0000-0003-2522-8338

Reviewer #1 (Public review): https://doi.org/10.7554/eLife.103191.4.sa1
Reviewer #2 (Public review): https://doi.org/10.7554/eLife.103191.4.sa2
Reviewer #3 (Public review): https://doi.org/10.7554/eLife.103191.4.sa3
Author response https://doi.org/10.7554/eLife.103191.4.sa4

## Additional files

### Supplementary files
MDAR checklist

### Data availability
Source data files containing summarized data have been provided for all figures. All raw (e.g., behavior videos and fluorescence microscopy images) and processed (e.g., animal and bacterial patch locations over time) data reported in this paper is publicly available at NDI Cloud (https://doi.org/10.63884/ndic.2025.pb77mj2s). All original code is publicly available at GitHub (https://github.com/shreklab/Haley-et-al-2024, copy archived at *shreklab, 2025*).

The following dataset was generated:

| Author(s) | Year | Dataset title | Dataset URL | Database and Identifier |
|---|---|---|---|---|
| Haley JA, Chen T, Aoi M, Chalasani SH | 2025 | Dataset: Accept-reject decision-making revealed via a quantitative and ethological study of *C. elegans* foraging | https://doi.org/10.63884/ndic.2025.pb77mj2s | NDI Cloud, 10.63884/ndic.2025.pb77mj2s |

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
