## [Editor Report · eLife Assessment]

Understanding how neural circuits mediate decision-making is a core problem in neuroscience. In this interesting and **important** work, the authors use detailed behavioral analysis and rigorous quantitative modeling to **convincingly** support the idea that the nematode *C. elegans* uses an "accept-reject" behavioral strategy, based on learned features of its environment, to make decisions upon encountering food patches. The work expands our understanding of the behavioral repertoire of this species, providing a foundation for future mechanistic studies in this powerful model system.

---

## [Referee Report · Reviewer #1 (Public review)]

Summary:

This work uses a novel, ethologically relevant behavioral task to explore decision-making paradigms in *C. elegans* foraging behavior. By rigorously quantifying multiple features of animal behavior as they navigate in a patch food environment, the authors provide strong evidence that worms exhibit one of three qualitatively distinct behavioral responses upon encountering a patch: (1) "search", in which the encountered patch is below the detection threshold; (2) "sample", in which animals detect a patch encounter and reduce their motor speed, but do not stay to exploit the resource and are therefore considered to have "rejected" it; and (3) "exploit", in which animals "accept" the patch and exploit the resource for tens of minutes. Interestingly, the probability of these outcomes varies with the density of the patch as well as the prior experience of the animal. Together, these experiments provide an interesting new framework for understanding the ability of the *C. elegans* nervous system to use sensory information and internal state to implement behavioral state decisions.

Strengths:

The work uses a novel, neuroethologically-inspired approach to studying foraging behavior

The studies are carried out with an exceptional level of quantitative rigor and attention to detail

Powerful quantitative modeling approaches including GLMs are used to study the behavioral states that worms enter upon encountering food, and the parameters that govern the decision about which state to enter

The work provides strong evidence that *C. elegans* can make 'accept-reject' decisions upon encountering a food resource

Accept-reject decisions depend on the quality of the food resource encountered as well as on internally represented features that provide measurements of multiple dimensions of internal state, including feeding status and time.

---

## [Referee Report · Reviewer #2 (Public review)]

This study provides an experimental and computational framework to examine and understand how *C. elegans* make decisions while foraging environments with patches of food. The authors show that *C. elegans* reject or accept food patches depending on a number of internal and external factors.

The key novelty of this paper is the explicit demonstration of behavior analysis and quantitative modeling to elucidate decision-making processes. In particular, the description of the exploring vs. exploiting phases, and sensing vs. non-sensing categories of foraging behavior based on the clustering of behavioral states defined in a multi-dimensional behavior-metrics space, and the implementation of a generalized linear model (GLM) whose parameters can provide quantitative biological interpretations.

The work builds on the literature of *C. elegans* foraging by adding the reject/accept framework.

---

## [Referee Report · Reviewer #3 (Public review)]

Summary:

In this study by Haley et al, the authors investigated explore-exploit foraging using *C. elegans* as a model system. Through an elegant set of patchy environment assays, the authors built a GLM based on past experience that predicts whether an animal will decide to stay on a patch to feed and exploit that resource, instead of choosing to leave and explore other patches.

Strengths:

I really enjoyed reading this paper. The experiments are simple and elegant, and address fundamental questions of foraging theory in a well-defined system. The experimental design is thoroughly vetted, and the authors provide a considerable volume of data to prove their points.

Weakness:

History-dependence of the GLM. The logistic GLM seems like a logical way to model a binary choice, and I think the parameters you chose are certainly important. However, the framing of them seem odd to me. I do not doubt the animals are assessing the current state of the patch with an assessment of past experience; that makes perfect logical sense. However, it seems odd to reduce past experience to the categories of recently exploited patch, recently encountered patch, and time since last exploitation. This implies the animals have some way of discriminating these past patch experiences and committing them to memory. Also, it seems logical that the time on these patches, not just their density, should also matter, just as the time without food matters. Time is inherent to memory. This model also imposes a prior categorization in trying to distinguish between sensed vs. not-sensed patches, which I criticized earlier. Only "sensed" patches are used in the model, but it is questionable whether worms genuinely do not "sense" these patches.

It seems more likely the worm simply has some memory of chemosensation and relative satiety, both of which increase on patches, and decrease while off of patches. The magnitudes are likely a function of patch density. That being said, I leave it up to the reader to decide how best to interpret the data.

Impact:

I think this work will have a solid impact on the field, as it provides tangible variables to test how animals assess their environment and decide to exploit resources. I think the strength of this research could be strengthened by a reassessment of their model that would both simplify it and provide testable timescales of satiety/starvation memory.

---

## [Author Response]

The following is the authors’ response to the previous reviews.

**Reviewer #1 (Public review):**
Summary:This work uses a novel, ethologically relevant behavioral task to explore decision-making paradigms in *C. elegans* foraging behavior. By rigorously quantifying multiple features of animal behavior as they navigate in a patch food environment, the authors provide strong evidence that worms exhibit one of three qualitatively distinct behavioral responses upon encountering a patch: (1) "search", in which the encountered patch is below the detection threshold; (2) "sample", in which animals detect a patch encounter and reduce their motor speed, but do not stay to exploit the resource and are therefore considered to have "rejected" it; and (3) "exploit", in which animals "accept" the patch and exploit the resource for tens of minutes. Interestingly, the probability of these outcomes varies with the density of the patch as well as the prior experience of the animal. Together, these experiments provide an interesting new framework for understanding the ability of the *C. elegans* nervous system to use sensory information and internal state to implement behavioral state decisions.Strengths:The work uses a novel, neuroethologically-inspired approach to studying foraging behaviorThe studies are carried out with an exceptional level of quantitative rigor and attention to detailPowerful quantitative modeling approaches including GLMs are used to study the behavioral states that worms enter upon encountering food, and the parameters that govern the decision about which state to enterThe work provides strong evidence that *C. elegans* can make 'accept-reject' decisions upon encountering a food resourceAccept-reject decisions depend on the quality of the food resource encountered as well as on internally represented features that provide measurements of multiple dimensions of internal state, including feeding status and time
**Reviewer #2 (Public review):**
This study provides an experimental and computational framework to examine and understand how *C. elegans* make decisions while foraging environments with patches of food. The authors show that *C. elegans* reject or accept food patches depending on a number of internal and external factors.The key novelty of this paper is the explicit demonstration of behavior analysis and quantitative modeling to elucidate decision-making processes. In particular, the description of the exploring vs. exploiting phases, and sensing vs. non-sensing categories of foraging behavior based on the clustering of behavioral states defined in a multi-dimensional behavior-metrics space, and the implementation of a generalized linear model (GLM) whose parameters can provide quantitative biological interpretations.The work builds on the literature of *C. elegans* foraging by adding the reject/accept framework.
**Reviewer #3 (Public review):**
Summary:In this study by Haley et al, the authors investigated explore-exploit foraging using *C. elegans* as a model system. Through an elegant set of patchy environment assays, the authors built a GLM based on past experience that predicts whether an animal will decide to stay on a patch to feed and exploit that resource, instead of choosing to leave and explore other patches.Strengths:I really enjoyed reading this paper. The experiments are simple and elegant, and address fundamental questions of foraging theory in a well-defined system. The experimental design is thoroughly vetted, and the authors provide a considerable volume of data to prove their points. My only criticisms have to do with the data interpretation, which I think are easily addressable.Weaknesses:History-dependence of the GLMThe logistic GLM seems like a logical way to model a binary choice, and I think the parameters you chose are certainly important. However, the framing of them seem odd to me. I do not doubt the animals are assessing the current state of the patch with an assessment of past experience; that makes perfect logical sense. However, it seems odd to reduce past experience to the categories of recently exploited patch, recently encountered patch, and time since last exploitation. This implies the animals have some way of discriminating these past patch experiences and committing them to memory. Also, it seems logical that the time on these patches, not just their density, should also matter, just as the time without food matters. Time is inherent to memory. This model also imposes a prior categorization in trying to distinguish between sensed vs. not-sensed patches, which I criticized earlier. Only "sensed" patches are used in the model, but it is questionable whether worms genuinely do not "sense" these patches.It seems more likely that the worm simply has some memory of chemosensation and relative satiety, both of which increase on patches and decrease while off of patches. The magnitudes are likely a function of patch density. That being said, I leave it up to the reader to decide how best to interpret the data.

Model design: We agree with the reviewer that past experience is not likely to be discretized into the exact parameters of our model. We have added to our manuscript to further clarify this point (lines 645-647). Investigating the mechanisms behind this behavior is beyond the scope of this project but is certainly an exciting trajectory for future *C. elegans* research.

osm-6The argument is that osm-6 animals can't sense food very well, so when they sense it, they enter the exploitation state by default. That is what they appear to do, but why? Clearly they are sensing the food in some other way, correct? Are ciliated neurons the only way worms can sense food? Don't they also actively pump on food, and can therefore sense the food entering their pharynx? I think you could provide further insight by commenting on this. Perhaps your decision model is dependent on comparing environmental sensing with pharyngeal sensing? Food intake certainly influences their decision, no? Perhaps food intake triggers exploitation behavior, which can be over-run by chemo/mechanosensory information?

osm-6 behavior: We thank the reviewer for pointing out the need to further elaborate on a mechanistic hypothesis to explain the behavior of osm-6 sensory mutants. We agree with the reviewer’s speculation that post-ingestive and other non-ciliary sensory cues likely drive detection of food. We have added additional commentary to our manuscript to state this (lines 529-538).

ImpactI think this work will have a solid impact on the field, as it provides tangible variables to test how animals assess their environment and decide to exploit resources. I think the strength of this research could be strengthened by a reassessment of their model that would both simplify it and provide testable timescales of satiety/starvation memory.
**Reviewer #2 (Recommendations for the authors):**
The authors have addressed most of my concerns.
**Reviewer #3 (Recommendations for the authors):**
The authors provide a considerable amount of processed data (great, thank you!), but it would be even better if they provided the raw data of the worm coordinates, and when and where these coordinates overlapped with patches. This is the raw data that was ultimately used for all the quantifications in the paper, and would be incredibly useful to readers who are interested in modeling the data themselves.This should not be prohibitive.

Data Availability: We thank the reviewer for pointing out this need. We are uploading all processed data (e.g. worm coordinates relative to the arena and patches) to a curated data storage server. We have updated our data availability statement to state this (lines 684-688).

Search vs. sample & sensing vs. non-sensing.The different definitions of behaviors in Figures 2H-K are a bit confusing. I think the confusion stems in part from the changing terms and color associations in Figures 2 H-K. Essentially the explore density in Figure 2 H is split into two densities based on the two densities (sensing vs. non-responding) observed in Figure 2I. In turn, the sensing density in Figure 2I is split into two densities (explore vs exploit) based on the two densities observed in Figure 2 H. But the way the figures are colored, yellow means search (Figure 2H) and non-responding (Figure 2I), green means exploit (Figure 2H) which includes sensing and non-responding, but also exclusively sensing (Figure 2I), and blue consistently means exploit in both figures. It might help to use two different color codes for Figures 2H and 2I, and then in 2J you define search as explore AND non-responding, sample as explore AND sensing, and exploit as exploit.

Color schema: While we understand the confusion, we believe that introducing additional colors may also present some misunderstandings. We have decided to leave the figure as it is.